# Centromere protection requires strict mitotic inactivation of the Bloom syndrome helicase complex

María Fernández-Casañas [1,4], Eleftheria Karanika[1,6], Umit Aliyaskarova [1,6], Tomisin Olukoga[1], Alex D. Herbert [1], Antony W. Oliver [2], Matthew Day [3], Adrijana Crncec [1,5] & Kok-Lung Chan [1] ✉

The BTRR (BLM/TOP3A/RMI1/RMI2) complex resolves DNA replication and recombination intermediates to maintain genome stability. Alongside PICH, they target mitotic DNA intertwinements, known as ultrafine DNA bridges, facilitating chromosome segregation. Both BLM and PICH undergo transient mitotic hyper-phosphorylation, but the biological significance of this remains elusive. Here, we uncover that during early mitosis, CDK1 and PLK1 constrain BTRR complex activities at centromeres. CDK1 destabilises the complex, limiting its binding to PICH at specialised chromatin underneath kinetochores. Inactivating the BLM-TOP3A interaction compromises the UFB-binding complex functions and prevents centromere destruction. Different phosphorylation on BLM affects the TRR subcomplex interaction and the mitotic activity, particularly phosphorylation at Ser144 and multiple PLK1-target sites suppresses illegitimate centromeric DNA unwinding. However, unleashing such activity after sister-chromatid cohesion inactivation facilitates the separation of entangled chromosomes. Here, we show a centromere protection pathway in human mitotic cells, heavily reliant on a tight spatiotemporal control of the BTRR complex.

Faithful chromosome segregation can be hindered by DNA replication and repair by-products, including double-stranded DNA catenanes, late replication intermediates and homologous recombination structures. These DNA interlinking molecules manifest as nucleosome-poor ultrafine DNA bridges (UFBs) that appear as cells disjoin sister chromatids during anaphase[1–5]. If not resolved properly, they can lead to chromosome mis-segregation, chromatin damage, chromothripsis, and gross chromosomal rearrangements – common characteristic of cancer cells[4–7]. Mammalian cells have evolved a multienzyme complex to specifically target anaphase UFBs, so-called the UFB-binding complex. Its key member is a SNF2-family DNA translocase, PICH (PLK1-

interacting checkpoint helicase; also known as ERCC6L)[2], which recruits the Bloom syndrome DNA helicase, BLM, and its interacting partners TOP3A (Topoisomerase 3A), RMI1 and RMI2 (RecQ-mediated genome instability protein complex 1 and 2; together known as the dissolvasome)[1,8]. Independently, PICH also recruits RIF1 (RAP1-interacting factor) on anaphase UFBs, but this interaction is suppressed by high levels of CDK1 before anaphase[9]. Apart from these core UFB-binding factors, Polo-like kinase 1 (PLK1) and Protein Phosphatases 1 (PP1s) are also found to associate with PICH and RIF1, respectively, but their biological roles remain unclear. Some components of the UFB-binding complex have important roles during DNA replication and

---

[1]Chromosome Dynamics and Stability Group, Genome Damage and Stability Centre, University of Sussex, Brighton, UK. [2]DNA Repair Enzymes Group, Genome Damage and Stability Centre, University of Sussex, Brighton, UK. [3]Centre for Molecular Cell Biology, School of Biological and Behavioural Sciences, Blizard Institute, Queen Mary University of London, London, UK. [4]Present address: Division of Cell and Molecular Biology, The Institute of Cancer Research, London, UK. [5]Present address: Laboratory of Cancer Biology and Genetics, Centre for Cancer Research, National Cancer Institute, Bethesda, MD, USA. [6]These authors contributed equally: Eleftheria Karanika, Umit Aliyaskarova. ✉e-mail: koklung.chan@sussex.ac.uk

repair, particularly the BTRR complex is crucial for suppressing excessive sister chromatid exchanges (SCEs)[10–12]. Mutations on BLM and the TRR (TOP3A/RMI1/RMI2) subcomplex have been linked to Bloom syndrome (BS) or BS-like genetic disorders, predisposing individuals to cancer development[13,14]. Given the vital role of these proteins as genome stability guardians, extensive studies have focused on understanding their cellular activities and regulation but mostly in S and G2 cells. Little is known about their role(s) during mitosis, except as core components of the UFB-binding complex.

Upon mitotic entry, both BLM helicase and PICH translocase undergo extensive phosphorylation mediated by several mitotic kinases, including CDK1/2, MPS1 and PLK1[2,15–20]. This is then followed by rapid dephosphorylation at the metaphase-anaphase transition. Over fifty mitosis-specific phosphorylation sites have been reported to occur on mitotic BLM, with the majority locating at the N-terminal half of the protein and outside the core helicase domain. The biological functions of most of these phosphorylation events remain unclear, but a relatively well-studied one is serine 144. It has been suggested that MPS1 phosphorylates BLM at serine 144, creating a docking site for the binding of PLK1 and the subsequent downstream phosphorylation, which is proposed to facilitate spindle assembly checkpoint (SAC) activation[16], yet the underlying mechanism remains undetermined. Besides, CDK1 can also phosphorylate BLM at multiple sites[15,18]. Recently, a study has suggested that serine 144 is also a target site of CDK1. In the presence of the binding of TOPBP1 (DNA Topoisomerase 2-binding protein 1) to BLM, it is believed that TOPBP1 can recruit additional PLK1 proteins to accelerate hyperphosphorylation of BLM during the G2-M transition. The lack of Ser144 or CDK1-mediated phosphorylation on BLM has been shown to abrogate BLM's anti-recombination activity in mitosis[18]. This leads to a proposal of mitotic phosphorylation-driven activation of BLM. Interestingly, the phosphorylated form of the BTRR complex has been reported to be expelled from mitotic chromosomes[21] while phosphorylation at threonine 182 of BLM can lead to Fbw7a-dependent polyubiquitylation and protein degradation in mitosis, albeit incomplete[22]. These findings generate a dilemma of how the phosphorylated BTRR complexes can effectively access DNA structures within the context of mitotic chromosomes. Nevertheless, another recent study has also suggested that mitotic phosphorylation of BLM is necessary for the assembly and activation of the UFB-binding complex[23]. Whether human cells attempt to further activate or deactivate the BTRR complex during mitosis remains controversial, but these studies highly indicate the existence of a regulatory control dependent on mitotic phosphorylation. Very strikingly, we and others have previously observed a serious mitotic catastrophe, triggered by BLM, when the PLK1 function is inhibited. Rather than the loss of BLM activity, in the absence of PLK1 the BTRR complex is rapidly recruited to the centromeres by PICH, driving severe centromere dechromatinisation and breakages[20,24]. The centromere damage was found to be initiated via active DNA unwinding by BLM at the chromatin layer underneath the kinetochores[20]. These findings raise several significant questions. Firstly, how does PLK1 protect the centromeres from being attacked by the BTRR dissolvasome? Does it do so by suppressing the BTRR complex activity, and/or by maintaining proper centromere compaction and conformation? Secondly, is the centromere protection reliant on mitotic phosphorylation of the BTRR complex and if so, what is the underlying mechanism?

In this work, we show that the BTRR complex is under a tight, transient inactivation by two key mitotic kinases, CDK1 and PLK1, upon the entry of mitosis until the onset of anaphase. This is crucial to prevent mis-assembly and activation of the UFB-binding complex at the kinetochore-associated chromatin, which otherwise causes unwanted centromere unwinding and destruction. High levels of CDK1 in early mitosis destabilise the BTRR complex and its binding to PICH, restricting the assembly of the UFB-binding complex. We then identify

critical regions and clusters of mitotic phosphorylation on BLM that affect the stability of the BTRR complex and its DNA unwinding ability at centromeres. Moreover, we demonstrate that the loss of phosphorylation at Ser144 and multiple PLK1-target sites can activate BLM DNA unwinding activity at centromeric chromatin, whereas the RIF1-PP1s axis is needed to counteract the CDK1's inhibitory effect on the assembly of the UFB-binding complex. Our study thus provides a molecular explanation of why the BTRR complex needs to be phosphorylated during a specific window of mitosis and, most importantly, it reveals a mitotic mechanism to safeguard centromere integrity in human cells. We generate a comprehensive model to discuss the significance of this centromere protection pathway in the maintenance of chromosome stability.

## Results

### Restriction of the BTRR complex activity before anaphase onset

During normal mitosis, both PICH and BLM are hyperphosphorylated by PLK1[2,16,20]. It is currently unknown whether and how this post-translational modification may contribute to the safeguarding of centromeres. Therefore, we started to investigate the role of PLK1 on regulating the mitotic activity of the BTRR complex. A previous study reported that mitotic phosphorylation can enhance BLM and PICH interaction to promote the formation of the UFB-binding complex[23]. However, inhibiting PLK1 with a specific small molecule, BI2536[25], did not impair the interaction of endogenous BLM and PICH proteins, despite the loss of protein hyperphosphorylation. Instead, the PLK1i treatments reproducibly enhanced BLM and PICH interaction in both prometaphase- and metaphase-arrested cells (Fig. 1a and Supplementary Fig. 1a). Inactivating PLK1 also slightly increased the interaction between BLM and TOP3A (Fig. 1a and Supplementary Fig. 1a). Moreover, although PICH was highly enriched at the kinetochore-associated chromatin (K-chromatin) on metaphase centromeres as shown by high-precision microscopy[20] and super-resolution STED (Supplementary Fig. 1b, c), we were unable to detect the binding of the BTRR complex and another PICH-interacting protein, RIF1, unless PLK1 was inhibited (Fig. 1b and Supplementary Fig. 1c). This suggests that the formation of the UFB-binding complex may be negatively regulated by PLK1 before anaphase onset.

To examine this phenomenon, we analysed the recruitment of the BTRR complex and RIF1 to UFB structures, but in pre-anaphase cells. RPE1 cells were depleted of SGO1 by RNA interference and treated with a low dose of TOP2 inhibitor, ICRF193, which led to premature loss of sister-chromatid cohesion and the accumulation of catenated UFBs in pre-anaphase cells, respectively (Fig. 1c and Supplementary Fig. 1d). We found that both the BTRR complex and RIF1[9] also poorly localised to the UFB structures that connect the prematurely separated sister chromatids in pre-anaphase cells (Fig. 1c and Supplementary Fig. 1e). In contrast, these factors were readily detected along DNA bridges in anaphase, particularly in cells progressing from the early to mid and late anaphase[9] (Fig. 1c and Supplementary Fig. 1e). In line with the above finding of the induction of K-chromatin localisation, acute inhibition of PLK1 triggered rapid loading of the BTRR complex onto the pre-anaphase UFBs and, crucially, induced DNA bridge unwinding as marked by RPA association (Fig. 1d). Moreover, this resulted in further separation of the catenated sister chromatids (Fig. 1d). Due to the persistent catenation along chromosomal arms (Fig. 1d; arrow), the distribution of split sister chromatids in the SGO1-depleted cells was generally disorganised and partial (Fig. 1e). However, this was converted into more anaphase-like patterns upon PLK1 inhibition, probably due to the activation of the UFB-binding complex (Fig. 1e). To rule out that the generation of anaphase-like cells might be caused by an unexpected escape to anaphase, we examined two well-known anaphase markers: Cyclin B1 degradation and midzone formation. Unlike the normal anaphase cells, we found that the anaphase-like populations

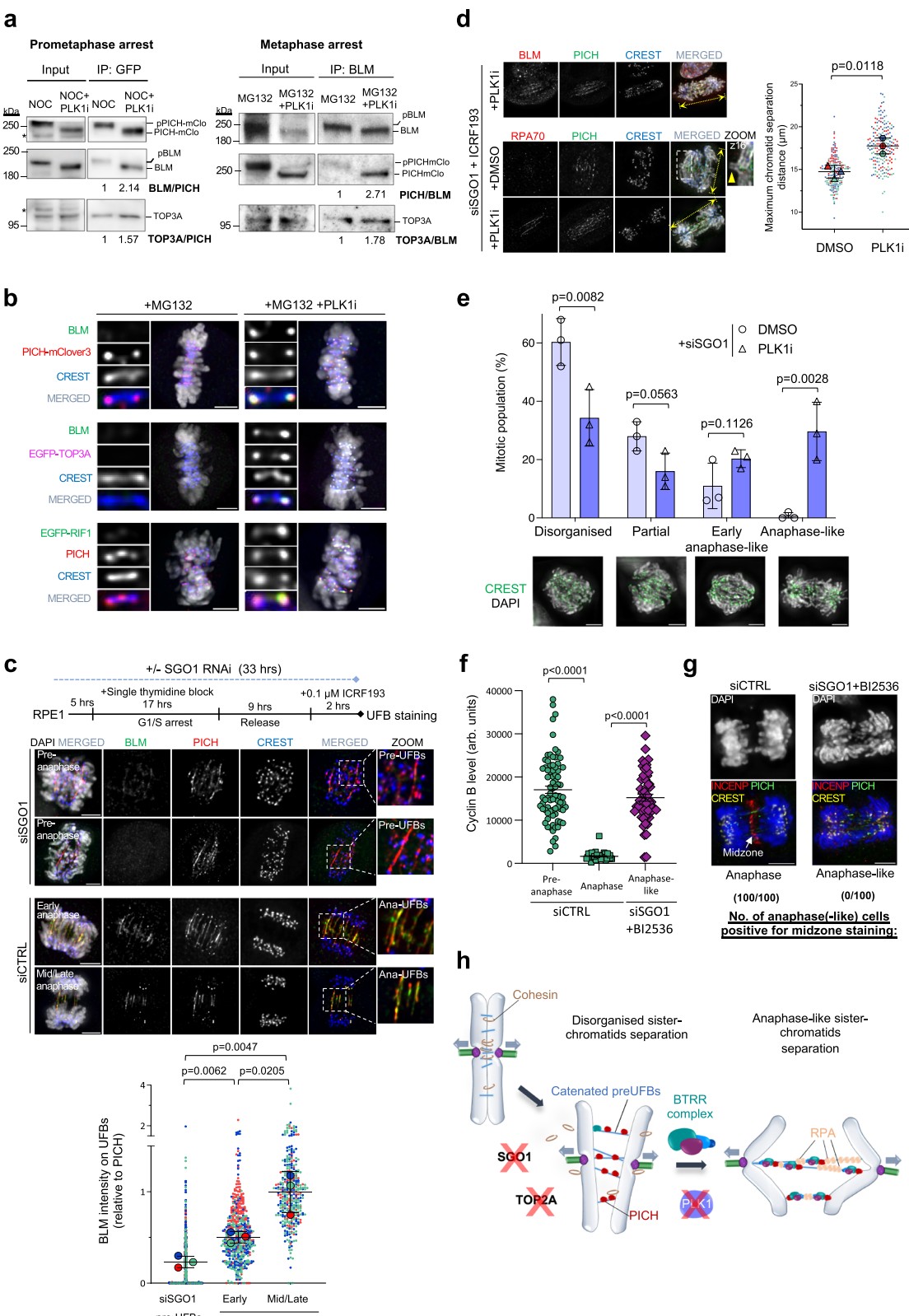

induced by PLK1 inhibition maintained high levels of Cyclin B1 and lacked midzone staining (Fig. 1f, g and Supplementary Fig. 1f), indicating that they remain at early mitosis. Therefore, the assembly of the UFB-binding complex is actively suppressed before anaphase onset in a PLK1-dependent manner, but its activation can facilitate chromosome separation after sister-chromatis cohesion loss (Fig. 1h).

## The assembly of functional UFB-binding complexes requires stable BTRR complex formation

To understand the underlying regulatory mechanism, we decided to determine the PICH-interacting domain(s) on BLM. Following systematic mutagenesis, we identified specific amino acids within the first thirteen residues of BLM that are crucial for proper association with PICH on anaphase UFBs (Fig. 2a, b and Supplementary Fig. 2a). Since

**Fig. 1 | Restricted activation of UFB-binding complexes before anaphase onset.**
**a** Co-immunoprecipitation in nocodazole/MG132-arrested RPE1 cells containing endogenous PICH-mAID-mClover3 ± PLK1i (100 nM). PLK1i was added at the last 1 hr. Relative IP ratios are shown. Repeated two and three experiments are shown in Supplementary Fig. 1a. **b** BLM, TOP3A and RIF1 at K-chromatin in MG132-arrested metaphase cells ± PLK1i (30 min). RPE1 endoPICH-mAID-mClover3 or HCT116 EGFP-TOP3A cells were co-stained for BLM; RPE1 EGFP-RIF1 cells were co-stained for PICH. Centromeres and K-chromatin are labelled by CREST and PICH, respectively.
**c** BLM UFBs localisation in pre-anaphase and anaphase cells. Experimental outline (top). Representative images of pre-anaphase UFBs in SGO1-depleted cells and UFBs in control cells. Relative BLM intensities are shown. Data is normalised to average intensities of mid/late anaphase-UFBs (total number of UFBs from three independent experiments: siSGO1 pre-UFBs n = 586, 78 cells; early anaphase UFBs n = 555, 43 cells; mid/late anaphase UFBs n = 343, 62 cells; mean ± S.D. is shown).
**d** similar in **c** but DMSO or PLK1i was added at the last 30 min. Representative images of BLM, PICH, and RPA on pre-anaphase UFBs. Maximum chromatid

separation distances indicated by the yellow lines are shown. Arrowhead, non-disjoined chromatid arms. Numbers of mitotic cells analysed from three independent experiments: DMSO n = 56, 61, 64; PLK1i: n = 63, 66, 66; mean ± S.D. is shown. **e** Mitotic populations showing different sister-chromatid separation patterns in ICRF193-treated SGO1-depleted RPE1 cells ± PLK1i (30 min). Representative images are shown below the graph. Numbers of cells analysed from three independent experiments: DMSO n = 56, 61, 64; PLK1i: n = 63, 66, 66; mean ± S.D. is shown. **f** The PLK1i-induced anaphase-like population maintained high Cyclin B1 (numbers of cells analysed: pre-anaphase n = 78, anaphase n = 42 and anaphase-like n = 52; mean ± S.E.M. is shown). **g** Absence of midzone in the anaphase-like cells. INCENP was used as a midzone marker. Numbers of cells examined shown below the images. **h** PLK1 inactivation promotes anaphase-like separation of catenated chromosomes after sister-chromatid cohesion inactivation. Scale bars, 5 µm. All p values are calculated by unpaired two-tailed t-test. Source data are provided as a Source data file.

the N-terminal region of BLM has been shown to interact with the TOP3A/RMI1/RMI2 (TRR) subcomplex[26,27], it is possible that the mutations introduced might disrupt the TRR subcomplex interaction. If this were the case, it would imply that either the TRR subcomplex mediates BLM-PICH interaction, or the formation of BTRR complexes is prerequisite for PICH association. However, since Sarlos et al. have shown that the binding of BLM to PICH does not require the TRR subcomplex[8], this led us to speculate that the very N-terminus of BLM may contain a PICH-interacting motif. Nonetheless, co-immunoprecipitation showed that most of the N-terminal mutants that did not localise to UFBs were highly defective in TOP3A pulldown (Fig. 2c). AlphaFold2 structural analyses also predicted that the first fifty residues of BLM, whose structure has not been elucidated, bind across both TOP3A and RMI1 via two consecutive interfaces. The residues from 35 to 50 have been experimentally shown for RMI1 interaction[28,29], while AlphaFold2 prediction suggested that an α-helix spanning residues 8 to 23 of BLM is responsible for TOP3A interaction (Fig. 2d and Supplementary Fig. 2b)[30,31]. Crucially, the point mutations we generated at Pro-5, Asn-8, Leu-9, and Gln-12, which are predicted to stabilise the helix or mediate hydrogen bonding with TOP3A, disrupted BLM-TOP3A interaction (Fig. 2c, d).

To strengthen the evidence that this is a bona fide TOP3A-interacting motif, we set up a cellular protein-protein interaction assay. EGFP-tagged wildtype, helicase-dead, or the N-terminal mutants of BLM were tethered to centromeres by fusion to the N-terminal domain of CENPB (residues 1–158)[32]. This serves as a "bait" to investigate TOP3A recruitment/interaction in vivo. Both the wildtype and helicase-dead BLM fusion proteins effectively recruited TOP3A to the centromeres, indicating the presence of BLM-TOP3A interaction, and this is independent of BLM helicase activity. Mutating the putative TOP3A-binding motif completely abolished the centromeric recruitment of TOP3A (Supplementary Fig. 2c). Collectively, this demonstrates that the binding of BLM to PICH on UFBs requires TOP3A interaction. Notably, in contrast to the BLM helicase-dead mutant (Q672R), which is only defective in DNA unwinding activity (i.e. the lack of centromeric RPA binding) but not K-chromatin localisation induced by PLK1i (Fig. 2e), the TOP3A-binding mutants were defective in both K-chromatin localisation and centromere unwinding/disintegration[20] (Fig. 2e, f). Since the loading of RPA at both K-chromatin and centromeres relies on active BLM, we used RPA foci/threads as an indicator of centromeric DNA unwinding throughout this study. These results indicate that the mitotic activity of the BTRR complex requires its stable formation, which is inconsistent with the previously proposed model[8]. To corroborate our findings, we depleted TOP3A in HeLa and U2OS cells and found this also abolished BLM localisation to UFBs in both cell lines (Supplementary Fig. 3a). Likewise, the ablation of BLM also prevented endogenous and ectopically expressed TOP3A from binding to PICH-coated UFBs (Supplementary Fig. 3b, c).

Therefore, we conclude that the assembly and activity of the UFB-binding complex in mitosis relies on intact BTRR complex formation.

## Disassembly of the mitotic BTRR complex at centromeres

Given this unexpected finding, we hypothesised that one potential mechanism to protect centromeres against the attack by the BTRR/UFB-binding complexes during mitosis could involve destabilisation of the BTRR complex. To test this, we combined the cellular protein-protein interaction assay and time-lapse microscopy to determine the interaction dynamics between different subunits of the BTRR complex in live cells. We tethered either mCherry-tagged RMI1 or helicase-dead BLM(Q672R) proteins to centromeres through the CENPB fusion and monitored the recruitment of EGFP-TOP3A protein throughout cell division in real-time (Fig. 3a). HeLa cells were chosen in these experiments because of the lesser disturbance of the cell cycle progression after the expression of the CENPB-fusion proteins. A helicase-dead BLM(Q672R) was also used to minimise potential interferences to centromeric structures. As planned, the RMI1 and BLM(Q672R) fusion proteins were successfully immobilised at centromeres in both interphase and mitotic cells (Fig. 3b). In the CENPB-mCherry-RMI1 expressing cells, EGFP-TOP3A co-localised with RMI1 at the centromeres from G2, throughout mitosis, to G1. There were no obvious changes of the TOP3A centromeric recruitment (Fig. 3b, upper panel; Supplementary Movie 1), indicating that the TOP3A-RMI1 interaction is constantly maintained, regardless of the cell cycle stages. In contrast, we detected a reduction of TOP3A recruitment to centromeres by the CENPB-mCherry-BLM(Q672R) fusion proteins when cells entered mitosis. Remarkably, the BLM-TOP3A interaction was quickly restored after anaphase onset (Fig. 3b, lower panel; Supplementary Movie 2). Similar changes of the interaction dynamics between the CENPB-mCherry-RMI1 and endogenous BLM proteins were also detected by quantitative immunofluorescence microscopy (Fig. 3c). The reduction of BLM binding to RMI1 in mitotic cells was also observed outside the centromere locus when RMI1 was tethered to telomeres via a fusion of the DNA-binding domain of Teb1 (TebDB)[33,34] (Supplementary Fig. 4a). These data support our prediction that the BTRR complex is destabilised, especially during early mitosis. However, in soluble protein extracts of mitotic cells, the formation of BTRR complexes remains detected[21,35]. Thus, our in vivo protein-protein interaction analysis may indicate that the stability of the BTRR complex is susceptible to chromatin environment and/or intrinsically weakened during mitosis. To further examine this, we performed co-immunoprecipitation of the BTRR complexes using interphase and mitotic cell extracts with washing buffers containing different salt concentrations. Repeatedly, we detected decreases of pull-down efficiencies between BLM to endogenous or GFP-tagged TOP3A proteins in mitotic extracts of different cell lines, particularly under high salt concentrations (Supplementary Fig. 4b, c). The unstable nature of the mitotic BTRR complex

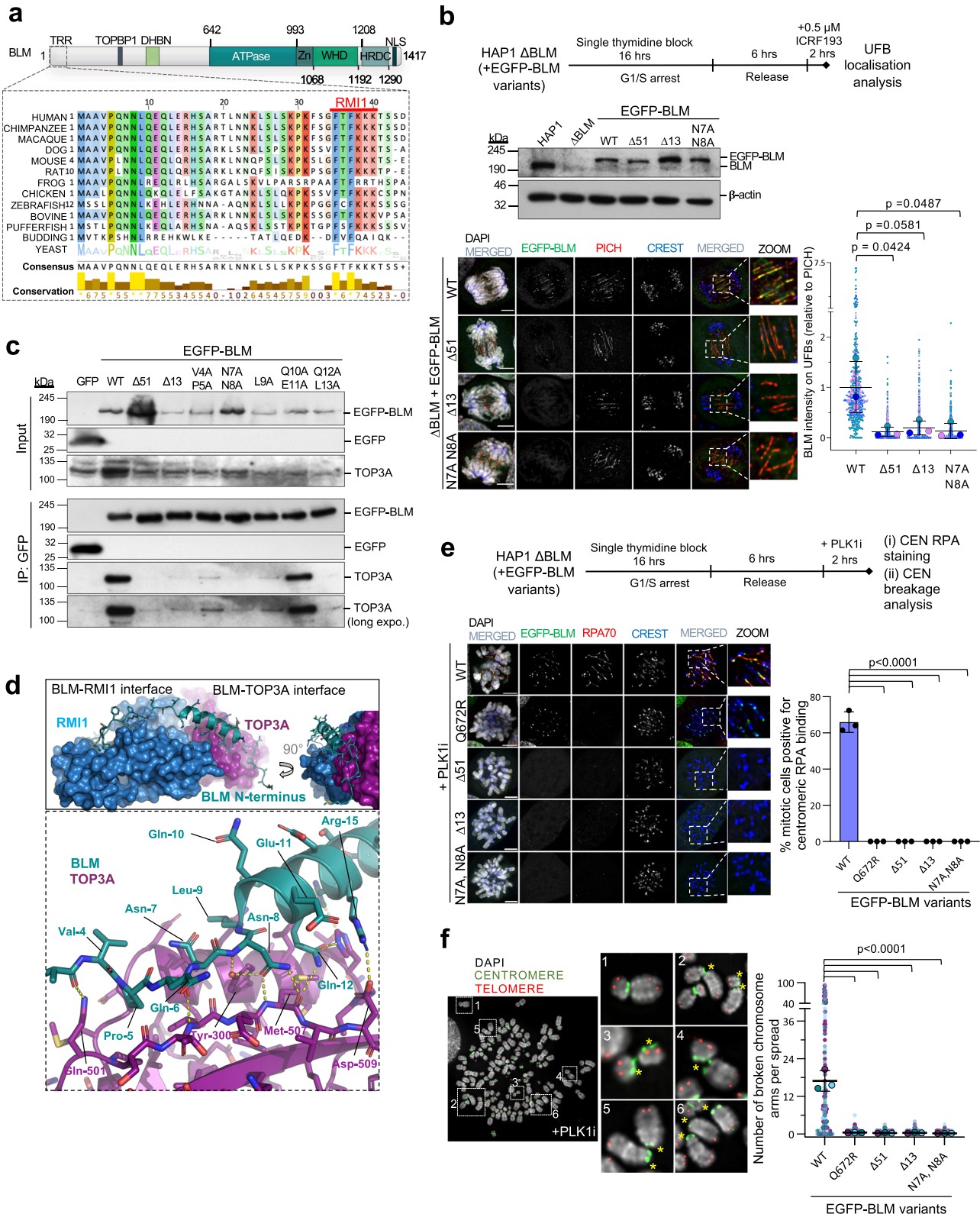

may hinder its proper association with PICH at the K-chromatin, thereby preventing mis-activation of the UFB-binding complex at centromeres.

## Mitotic phosphorylation weakens the BLM and the TRR-subcomplex interacting surfaces

Several mitotic kinases have been shown to phosphorylate the subunits of the BTRR complex[15,16,19,20]. We next tested whether they may

involve in destabilising the complex during early mitosis. The CENPB-mCherry-RMI1 metaphase-arrested cells were treated with different specific inhibitors targeting PLK1, CDK1, MPS1, Aurora A, or Aurora B. A short treatment time was applied to avoid triggering premature mitotic exit, particularly during CDK1 and MPS1 inhibition[36–38]. We found that inhibition of PLK1 or CDK1 can effectively restore the binding of BLM to the centromere-tethered RMI1, while Aurora B inhibition exhibited a partially weak effect (Supplementary Fig. 5). By

**Fig. 2 | Disruption of BLM and TOP3A interaction impairs UFB-binding complex activities. a** Sequence comparison of the N-terminus of BLM among different species. RMI1-binding motif (red) (TRR: TOP3A/RMI1/RMI2-binding regions; TPB: TOPBP1-binding site; DHBN; dimerization domain; WHD: Winged-Helix domain; HRDC: Helicase-and-RNaseD C-terminal domain; NLS: Nuclear localisation signal). **b** The N-terminal truncations or mutations impair BLM UFB localisation. Experimental setup is shown (top). Western Blot shows BLM levels (β-actin: loading control). Representative images showing BLM localisation on UFBs labelled by PICH. Relative BLM intensities on UFBs (normalised to average intensities of WT EGFP-BLM) are shown. Numbers of UFBs measured from three independent experiments: WT $n = 340, 113, 88$; Δ51 $n = 373, 126, 86$; Δ13 $n = 306, 135, 68$; N7A N8A $n = 312, 101, 71$; mean ± S.D. is shown; Scale bars, 5 μm. **c** The BLM mutations impair TOP3A interaction. EGFP-tagged BLM variants were transiently expressed in HEK293T cells for co-immunoprecipitation. One co-IP experiment was performed using the indicated BLM variants. **d** AlphaFold2 predicts the N-terminus of BLM

(cartoon) interacting with TOP3A and RMI1 (surface). The interacting residues are shown as sticks, with hydrogen bonds (dashed lines). **e** BLM mutations impair PLK1i-induced K-chromatin localisation and centromere unwinding. Experimental setup (top). Representative images of ΔBLM HAP1 mitotic cells complemented with EGFP-tagged BLM variants treated with 100 nM BI2536. Percentages of mitotic cells positive for centromere unwinding are shown. Total number of cells analysed from three independent experiments: WT $n = 269$; Q672R $n = 182$; Δ51 $n = 179$; Δ13 $n = 183$; N7A, N8A $n = 184$; mean ± S.D. is shown; Scale bars, 5 μm. **f** similar to **e**, BLM mutations impair PLK1i-induced centromere disintegration. Representative images of centromere breakages (asterisks). Quantification of broken whole chromosome arms per spread after 60 nM PLK1i treatments. Numbers of mitotic spreads analysed from three independent experiments: WT $n = 81, 52, 46$; Q672R $n = 82, 33, 32$; Δ51 $n = 79, 34, 30$; Δ13 $n = 84, 33, 31$; N7A, N8A $n = 79, 35, 30$; mean ± S.D. is shown. All $p$ values are calculated by unpaired two-tailed $t$-test. Source data are provided as a Source data file.

integrating data obtained from mass spectrometry analyses[30,39–42] and our AlphaFold2 prediction, we speculated that mitotic phosphorylation at six potential serine or threonine sites on the N-terminus of BLM (Ser17, Thr20, Ser26, Ser28, Ser33 and Thr36) and the one at the C-terminal end (Ser1417) might disrupt the TOP3A- and RMI1-binding interfaces, affecting stable interaction between BLM and the TRR subcomplex (Fig. 3d). As predicted, mutations at either all seven residues or the six residues at the N-terminus of BLM successfully rescued the recruitment of EGFP-TOP3A to centromeres by the CENPB-tagged BLM fusion proteins in early mitotic cells (Fig. 3e). Furthermore, we found that double mutations at Ser17 and Thr20 in the N-terminal helix of BLM were also sufficient to restore the centromeric recruitment of TOP3A (Fig. 3e). These results indicate a phosphorylation-dependent mechanism to destabilise the BTRR complex assembly during mitosis.

## CDK1, MPS1 and PLK1 contribute differently to suppress the UFB-binding complex

Next, we tested whether stabilising mitotic assembly of the BTRR complex was able to promote its binding to PICH at K-chromatin. HAP1 ΔBLM cells and RPE1 cells complemented with EGFP-tagged BLM were arrested in metaphase and treated with different kinase inhibitors (Fig. 4a and Supplementary Fig. 6a). PLK1 inhibition, as expected, effectively induced the illegitimate loading of BTRR complexes onto K-chromatin. We found that inhibition of CDK1, which was shown to enhance mitotic BTRR complex assembly, also induces the K-chromatin localisation (Fig. 4b and Supplementary Fig. 6a). However, intriguingly, unlike PLK1 inhibition, the CDK1i treatments poorly triggers centromeric DNA unwinding (Fig. 4c). This data indicates that CDK1 and PLK1 play different roles to regulate mitotic BTRR complexes. Thus, PLK1 has an additional function to suppress BLM-mediated centromeric DNA unwinding. Alternatively, PLK1 may also act to prevent centromeric DNA structures from being misrecognised as a BLM substrate.

To dissect these possibilities, we studied the function of mitotic phosphorylation on BLM. It has been proposed that MPS1 can catalyse a priming phosphorylation at serine 144 of BLM to promote PLK1 docking and subsequent hyper-phosphorylation[16]. If the mitotic BLM activity can be negatively influenced by PLK1-mediated phosphorylation, we suspected that inactivating MPS1 might unleash BLM centromere unwinding once the BTRR complex gained the access to K-chromatin via CDK1 inhibition. Indeed, acute co-treatments of both CDK1 and MPS1 inhibitors induced single-stranded DNA formation at K-chromatin, despite less efficient than the PLK1i alone (Fig. 4c). To further test whether the suppression of mitotic BLM DNA unwinding activity is attributed to S144 phosphorylation, we stably expressed wildtype and S144A phosphomutant BLM in HAP1 ΔBLM cells and treated them with PLK1 or CDK1 inhibitors. The S144A phosphomutant, like wildtype BLM, was unable to localise to K-chromatin unless PLK1 or

CDK1 was inhibited. More importantly, upon the CDK1 inhibition, mitotic cells expressing the S144A phosphomutant exhibited a six-fold increase in centromeric DNA unwinding as compared to the wildtype counterpart (Fig. 4d). This indicates that the S144A BLM mutant is dominantly active. It was worth noting that the elevation of centromeric DNA unwinding was greater in S144A cells than in the wildtype cells treated with MPS1i (Fig. 4c, d). This might be caused by incomplete MPS1 inhibition because of the restricted treatment time, and/or Ser144 can be targeted by another mitotic kinase(s). To further test the effect of S144 phosphorylation on mitotic BLM suppression, we generated a phospho-mimicking BLM mutant, S144E. Interestingly, we observed partial suppression of the centromere unwinding activity (Fig. 4d). This may suggest that glutamic acid substitution was not fully mimicking the function of a phosphate group. Nevertheless, our data is somewhat inconsistent with a recent study in HeLa cells claiming that S144 phosphorylation is required for promoting mitotic BLM activity to suppress excessive crossovers[18]. We thus measured SCE frequencies in S144A mutant HAP1 cells but did not detect any obvious defects in SCEs suppression (Supplementary Fig. 6b). Our results were indeed consistent with another report examining S144A-complemented Bloom's Syndrome cells showing no defects in anti-hyperrecombination[16]. Whether this is a cell line-specific phenomenon remains unclear, but it is worth noting that the induction of BLM-mediated centromere unwinding by PLK1 inhibition was also observed in HeLa cells[20]. Therefore, we conclude that phosphorylation of serine 144 functions to restrain, rather than promote, BLM activity during mitosis.

To further confirm that the mitotic suppression of BLM relies on the downstream PLK1 phosphorylation, we performed mass spectrometry analyses and comprehensive phosphoproteomic comparison on BLM[39,41,43–47], and identified twelve phosphorylated serine/threonine sites that are highly dependent on PLK1 activity. Interestingly, three phosphorylation events were particularly observed upon PLK1 inhibition (Supplementary Fig. 6c). We thus complemented HAP1 ΔBLM cells with the BLM phosphomutants that abolish the twelve putative PLK1 sites either with or without the three additional phospho-mimicking glutamate substitutions (12A3E or 12 A) (Fig. 4e, f). Consistent with the findings in the S144A mutants, both 12A3E or 12A mutations led to significant increases in K-chromatin unwinding upon CDK1 inhibition (Fig. 4f). Next, we focused on a small cluster of the putative PLK1 sites (Ser282, Ser336, Ser337, Ser338, Ser358 and Ser367) located within the central region of the N-terminal half of BLM. Previous in vitro kinase assays have shown that this region is targeted by recombinant PLK1[16]. Besides, this region also contains domains for TOPBP1-interaction[29,48] and BLM dimerization[49], respectively, which made us to speculate that phosphorylation surrounding them might influence BLM activity. Indeed, we found that mutations of the six serine residues (6A-1) within this region also

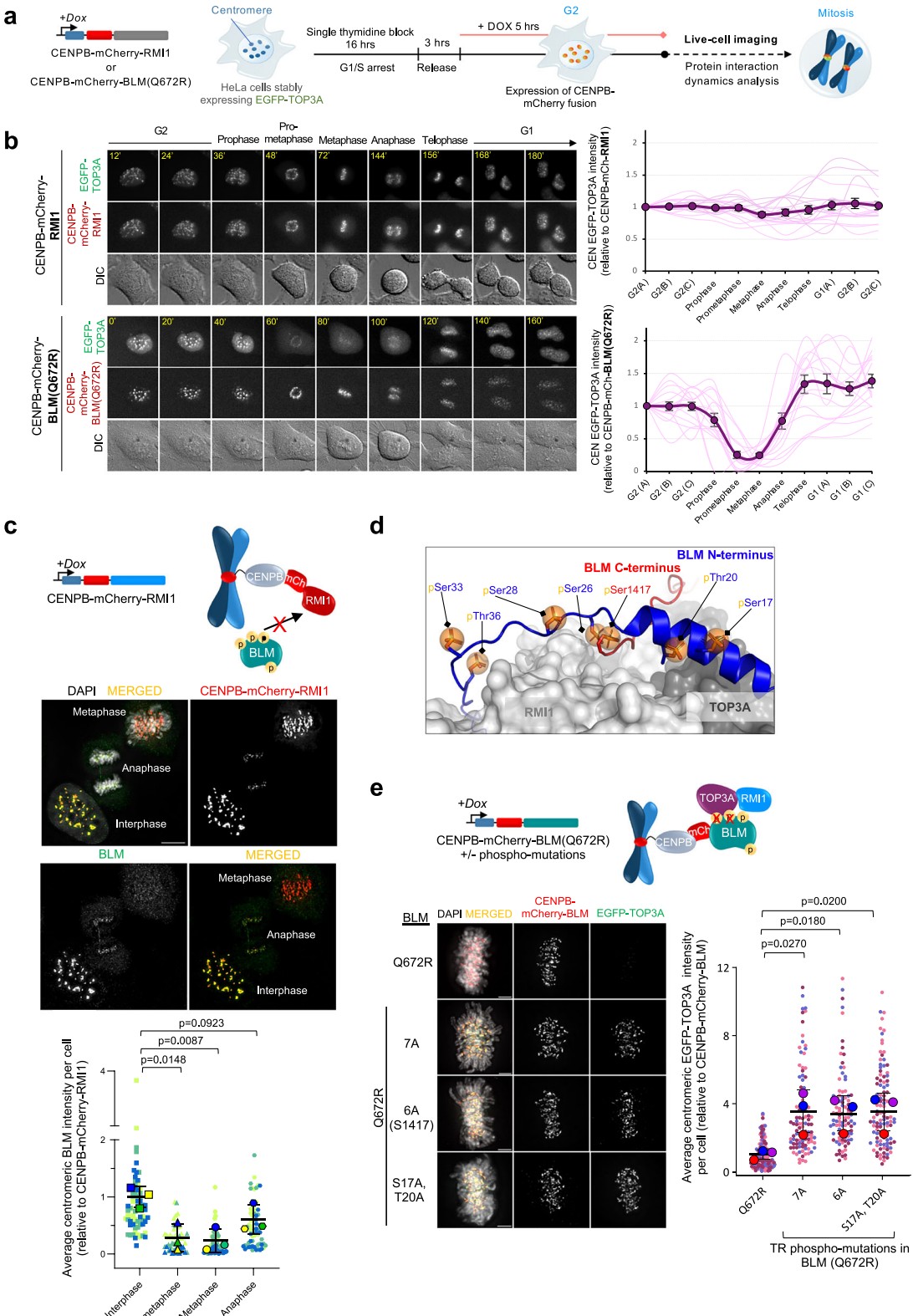

elevate centromere unwinding upon CDK1 inhibition (Fig. 4e, f). However, interestingly, mutations at the other six putative PLK1 sites (6A-2) conferred a similar activation effect, albeit less effective (Fig. 4e, f). Therefore, in order to fully suppress the mitotic activity, phosphorylation at multiple sites or regions of BLM is probably required, which could explain why BLM is highly phosphorylated during early mitosis.

## PLK1 and CDK1/Cyclin B1 directly phosphorylate the N-terminal fragments of BLM in vitro

Our cellular analysis indicates that the restraint of mitotic BLM activity relies on PLK1-dependent phosphorylation. To test whether the proposed residues on BLM can be directly targeted by PLK1, recombinant BLM protein fragments consisting of residues 1 to 83 or 272 to 552, which cover all the twelve putative PLK1 sites, were expressed and

**Fig. 3 | Dynamic changes of the BLM-TOP3A interaction at centromeres during mitosis. a** In vivo live-cell imaging assays measuring the dynamics of protein-protein interaction at centromeres in EGFP-TOP3A HeLa cells expressing either CENPB-mCherry-RM1 or CENPB-mCherry-BLM(Q672R) fusion proteins. **b** Time-lapse live-cell images showing the TOP3A-RMI1 (upper panel) and TOP3A-BLM interaction (lower panel) at centromeres from late G2 to the next G1. Right: Quantification of centromeric EGFP-TOP3A intensities (relative to CENPB-mCherry-RMI1 or CENPB-mCherry-BLM(Q672R)) in the indicated cell-cycle stages (number of cells analysed: CENPB-mCherry-RMI1 $n = 16$ from a single experiment; CENPB-mCherry-BLM(Q672R) $n = 16$ from two independent experiments; mean ± S.E.M is shown). **c** Quantitative immunofluorescence microscopy showing the recruitment of endogenous BLM to centromeres by CENPB-mCherry-RMI1 in HeLa interphase and mitotic cells. Right: Quantification of average centromeric BLM intensity relative to CENPB-mCherry-RMI1 per cell. Data is normalised to the average intensity in metaphase (total numbers of cells analysed: Interphase $n = 70$; Prometaphase $n = 51$; Metaphase $n = 69$, Anaphase $n = 66$ from three independent experiments; mean ± S.D. is shown). Scales bars, 10 μm. **d** A diagram shows the mitotic phosphorylation sites in the N- and C-termini of BLM that span the interacting surfaces for TOP3A and RMI1. AlphaFold2 model illustrates the BLM as a helix in cartoon representation rainbow coloured from N-terminus (blue) to C-terminus (red), while TOP3A and RMI1 are depicted as surfaces. Phosphorylated residues on BLM are represented as spheres. **e** Immunofluorescence images showing centromeric co-localisation of EGFP-TOP3A and CENPB-mCherry-BLM(Q672R) with or without the phosphorylation mutations in metaphase HeLa cells. (7 A: S17, T20, S26, S28, S33, T36 and S1417 converted into alanine; 6A(S1417): except S1417A mutation). Quantification of average centromeric EGFP-TOP3A intensity relative to CENPB-mCherry-BLM(Q672R) variants per cell is shown. (numbers of cells analysed: Q672R alone $n = 117$; 7A $n = 99$; 6 A(S1417) $n = 97$; S17A, T20A $n = 106$ from three independent experiments. mean ± S.D. is shown). All $p$ values are calculated by unpaired two-tailed $t$-test. Source data are provided as a Source data file.

purified in *E.coli*, and subject to in vitro PLK1 kinase reaction and mass spectrometry phosphorylation analysis (Fig. 5a–d). We found that eight out of the twelve putative PLK1 sites were significantly phosphorylated by PLK1 in vitro (Fig. 4e, c, d; red-coloured residues). Phosphorylation at two other proposed sites, Thr50 and Ser336, was also detected, albeit with lower confidence. In addition, we also mapped extra PLK1 phosphorylation sites that were previously noticed in mitotic cells (Fig. 5c, d; black-colour residues). Two unreported PLK1 phosphorylation sites, Ser482 and Ser517, were also identified. Therefore, in total at least 22 residues spanning the N-terminus of BLM can be directly phosphorylated by PLK1 in vitro. We thus concluded that BLM is a bona fide substrate of PLK1. To facilitate the PLK1-mediated phosphorylation in vivo, it has been claimed to rely on a priming phosphorylation at Ser144 by MPS1[16]. However, we were unable to detect phosphorylation at Ser144 on a BLM fragment comprised of residues 121 to 166 even after prolonged incubation with recombinant MPS1 in vitro (Fig. 5e). This was not due to the lack of MPS1 activity because phosphorylation was readily detected in a control peptide containing known MPS1/PLK1 sites[50] (Fig. 5f). Alternatively, the short BLM fragments (121-166) might not be an optimal substrate. However, we found that it can be directly phosphorylated by recombinant CDK1/CyclinB1 (Fig. 5e), and more interestingly, the target site was mapped at Ser144 (Fig. 5g). Therefore, we believe that CDK1, but not MPS1, is more likely to be responsible for the S144 phosphorylation in vivo. Together with the finding of the weak activation of BLM by MPS1 inhibition, this suggests that MPS1 probably plays a rather indirect role in the regulation of BLM during mitosis.

## RIF1-PP1s are an auxiliary factor for the UFB-binding complex activation before anaphase

Thus far, we have demonstrated that PLK1 acts as a major mitotic suppressor of the BTRR/UFB-binding complexes. However, unexpectedly, we found that the activation of the complexes through the PLK1i treatments becomes less efficient in cells lacking RIF1. Not only the reduction of centromere unwinding, but also the assembly at K-chromatin were partially impaired (Fig. 6a). One possible explanation might be that the PLK1-mediated inhibitory phosphorylation cannot be removed timely in the absence of RIF1. However, gel shift analyses revealed that, regardless of RIF1, both BLM and PICH proteins lost hyperphosphorylation after PLK1i treatments (Supplementary Fig. 7a, asterisks). Intriguingly, inhibiting CDK1 in the ΔRIF1 cells remained effective to trigger K-chromatin localisation of the BTRR/UFB-binding complexes (Fig. 6b). Therefore, CDK1, but not PLK1, is primarily responsible for limiting the complex assembly, consistent with our data shown above. RIF1 has been shown to interact and recruit Protein Phosphatases 1 (PP1s) to PICH[23]. We thus hypothesised that during PLK1 inactivation, this might lead to the restoration of RIF1-PP1s interaction and/or PP1s activity, which helps counteract the CDK1-dependent inhibitory phosphorylation and relieve the suppression of

the protein complex assembly. As such, the binding between RIF1 and PP1s, coupled with PP1s activity, is likely to be critical. Supporting this notion, we found that centromere unwinding and breakages were less efficiently induced in RPE1 ΔRIF1 cells complemented with a PP1-binding mutant of RIF1 (PP1bs)[51] (Fig. 6c, d and Supplementary Fig. 7b). Moreover, treating cells with a specific inhibitor, tautomycetin[52], targeting PP1s enzymatic activity also hindered the binding of the BTRR complexes to K-chromatin induced by PLK1 inhibition (Fig. 6e). Collectively, these results demonstrate that the regulation of spatio-temporal activity of the BTRR and UFB-binding complexes requires cooperative actions of CDK1, PLK1 and RIF1-PP1s.

## Discussion

Considering the brief duration of anaphase, it is reasonable to believe that the UFB-binding complex factors may have been activated prior to anaphase onset, which, in principle, can facilitate rapid separation of sister chromatids thereafter[18,23]. Contrary to this concept, our current study reveals that human cells actively constrain the assembly and spatiotemporal activity of the UFB-binding complex before chromosome segregation. This suppression is vital to prevent illegitimate DNA unwinding at the kinetochore-associated chromatin (K-chromatin): an initiation step that leads to centromere disintegration and hence failure of chromosome alignment[20,24]. To avoid such mitotic catastrophe, we show that high levels of CDK1 in mitotic cells destabilise the BTRR dissolvasome formation, which prevents its stable association with PICH at K-chromatin and hence restricts the dissolvasome to access the centromeres. Concurrently, PLK1 mediates additional phosphorylation at multiple sites on BLM after docking at phosphorylated S144[16], to suppress illegitimate DNA unwinding at the centromeres to protect their integrity (Fig. 7a). It was assumed that phosphorylation of S144 is initiated by MPS1[16] but our in vitro kinase experiments suggest that this is mediated by CDK1 instead. Our finding also aligns well with the previous research showing very low substrate specificity of the S144 sequence for MPS1[53]. However, we cannot rule out that MPS1 may play an indirect role to facilitate S144 phosphorylation in vivo. Nevertheless, the two-tier regulatory system, through destabilising the BTRR/UFB-binding complex formation and suppressing BLM DNA unwinding activity, offers several advantages. Firstly, it prevents the BTRR complex from interfering with chromatin regions where only PICH activity is required, such as at the K-chromatin/centromeres. Secondly, the suppression of BLM activity minimises unwanted chromatin remodelling, even if the complex is accidentally misplaced. Additionally, because PICH can still associate with UFB precursors, rapid assembly of the UFB-binding complex can be achieved upon the removal of the inhibitory phosphorylation at the metaphase-anaphase transition. This swift process is highly beneficial for timely segregation of chromosomes. Indeed, we have demonstrated that acute PLK1 inhibition, following the inactivation of sister chromatid cohesion, can

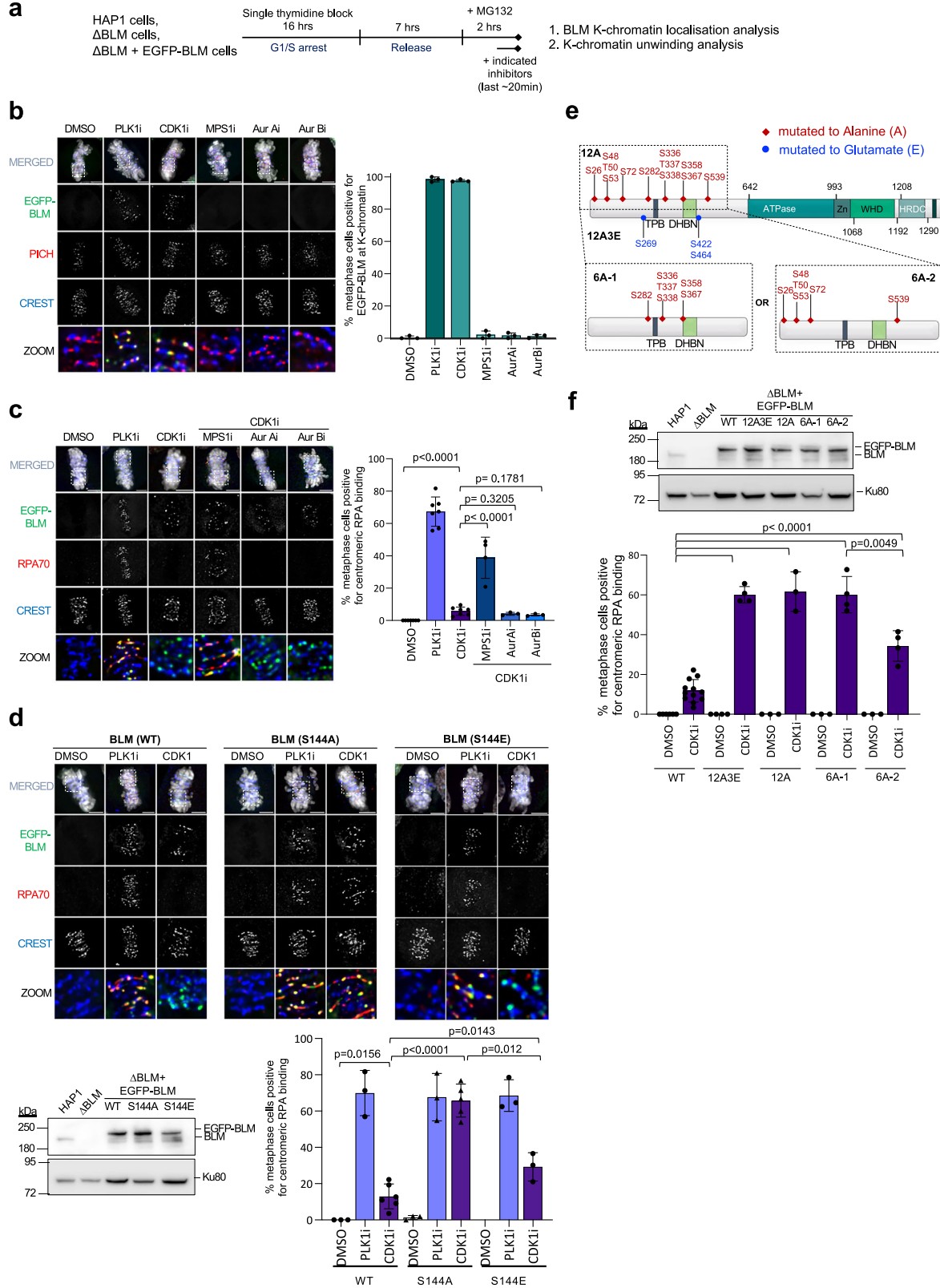

promote the separation of catenated chromosomes, mimicking anaphase chromosome segregation.

In the current study, we have shown that CDK1 can block the access of BTRR complexes to K-chromatin. Besides, it probably also phosphorylates BLM at S144 to promote the downstream hyperphosphorylation mediated by PLK1. If CDK1 can play such a master role in suppressing both the assembly and activity of the BTRR/UFB-binding

complexes, then why was CDK1 inhibition alone not efficient to trigger centromere unwinding? One plausible explanation is that in our mitotic cellular assays, cells require to fully commit into mitosis before CDK1 inactivation. Under such conditions, most of the components of the UFB-binding complexes should have been fully phosphorylated. The subsequent short CDK1i treatments (maximum of 20 min to avoid premature mitotic exit) may not lead to complete removal of the

**Fig. 4 | CDK1, MPS1 and PLK1 contribute differently to suppress the UFB-binding complexes. a** Experimental setup. **b** CDK1 or PLK1 inhibition induces BLM localisation to K-chromatin. Representative images of the metaphase-arrested cells treated with the indicated kinase inhibitors. PICH labels for K-chromatin, CREST for centromeres. Quantification of cells positive for BLM at K-chromatin. Numbers of metaphase cells analysed from three independent experiments: DMSO $n$ = 54, 62, 59; PLK1i $n$ = 63, 66, 72; CDK1i $n$ = 70, 74, 66; MPS1i $n$ = 65, 56, 49; AurAi $n$ = 63, 59, 55; AurBi $n$ = 56, 52, 46.**c** Kinase inhibitors co-treatments. Images and quantification of cells positive for RPA threads/foci at >5 centromeres. Numbers of metaphase analysed from at least three independent experiments: DMSO $n$ = 56, 56, 56, 69, 75, 49, 45; PLK1i $n$ = 52, 60, 65, 80, 72, 68, 79; CDK1i $n$ = 45, 85, 66, 73, 60, 75, 44; CDK1i/MPS1i $n$ = 43, 60, 72, 77; CDK1i/AurAi $n$ = 80, 63, 73; CDK1i/AurBi $n$ = 59, 48, 71. **d** Representative images K-chromatin localisation and centromeric RPA of EGFP-tagged BLM variants. Western blotting of BLM. Quantification of metaphase

positive for centromeric RPA threads/foci. Numbers of metaphase analysed from at least three independent experiments: EGFP-BLM WT DMSO $n$ = 72, 56, 69; PLK1i $n$ = 60, 65, 80; CDK1i $n$ = 85, 66, 73, 83, 90, 55; S144A DMSO $n$ = 48, 53, 64; PLK1i $n$ = 62, 63, 58; CDK1i $n$ = 65, 60, 63, 95, 45; S144E DMSO $n$ = 53, 40, 42; PLK1i $n$ = 56, 65, 72; CDK1i $n$ = 53, 76, 52. **e** BLM residues mutated to alanine or glutamate. **f** Western blotting of BLM (top). Quantification of metaphase positive for centromeric RPA threads/foci. Numbers of metaphase analysed from at least three independent experiments: EGFP-BLM WT: DMSO $n$ = 58, 64, 59, 62, 52, 53; CDK1i $n$ = 53, 61, 70, 56, 79, 52, 57, 36, 55, 50, 45, 36; 12 A3E: DMSO $n$ = 60, 58, 66, 53; CDK1i $n$ = 86, 93, 100, 84; 12A: DMSO $n$ = 72, 70, 67; CDK1i $n$ = 81, 112, 96; 6A-1: DMSO $n$ = 67, 80, 54; CDK1i $n$ = 50, 87, 81, 44; 6A-2: DMSO $n$ = 55, 48, 65; CDK1i $n$ = 82, 65, 60, 50. Scale bars, 5 μm. Mean ± S.D. is shown in all graphs. All $p$ values are calculated by unpaired two-tailed $t$-test. Source data are provided as a Source data file.

inhibitory phosphorylation. Therefore, an efficient activation of the BTRR complexes may not achieve. Besides, active PLK1 may suppress the phosphatases and delay protein dephosphorylation. Supporting this notion, we show that the lack of RIF1 and PP1s activity can impair efficient activation of the UFB-binding complex (Fig. 7b).

One of the most important findings of our study is the demonstration of a mitotic mechanism for centromere protection, which relies on the strict spatiotemporal regulation of the BTRR/UFB-binding complexes by CDK1 and PLK1. It is worth noting that both kinases have also been shown to suppress components of the non-homologous end-joining (NHEJ) and homologous recombination (HR) machineries including 53BP1 and RAD51[54,55], which lead to their active exclusion from damaged mitotic chromosomes[56–58]. The mitotic suppression of 53BP1 recruitment was shown to be mediated by specific phosphorylation at T1609 and S1618 sites that block the binding of 53BP1 to ubiquitinated histones[59,60]. Overriding such suppression by expressing 53BP1 phosphomutants has been shown to impair faithful chromosome segregation[59]. Similarly, our current research reveals that in addition to the shut-down of the major DNA repair machineries, programmed inactivation of chromatin remodelling/processing factors through mitotic phosphorylation is essential to specifically protect the centromere structures during mitosis.

The centromere protection pathway described here is governed by several important factors: (1) the integrity of the mitotic BTRR complex; (2) the interaction between the dissolvasome and PICH; (3) the activity of BLM helicase; and (4) the action of mitotic kinases and protein phosphatases. It is generally accepted that the BTRR complexes form stably, irrespective of the cell cycle stage[18,21,26,61,62]. Thus, we were surprised to observe its dynamic assembly and disassembly on mitotic chromatin. Typically, most protein-protein interaction analyses are performed under the elimination of DNA/chromatin to avoid indirect co-immunoprecipitation, which may have overlooked its biological influence on protein complex formation. Our live-cell imaging approach provides an alternative method to study protein-protein interactions under physiological conditions. An important implication of our findings is that chromatin, and its associated factors, likely exert a significant impact on the stability of DNA-binding complexes. Although we also detected the destabilisation of BTRR complex formation outside the centromeric chromatin, the fact that centromeres and kinetochores can accommodate numerous protein kinases[63–65], these highly localised kinase activities may potentially further destabilise the BTRR/UFB-binding complexes. Interestingly, we occasionally observe weak BLM localisation on the protruded pre-anaphase UFBs in the SGO1-delpeted cells, which are distanced from the core centromeres (the zone of high kinase activity). This could be attributed to a partial relief from the phosphorylation-dependent inhibition.

The mitotic phosphorylation has been previously proposed to stabilise the interaction between PICH and the BTRR complex, as well as to stimulate the DNA unwinding activity on UFBs[23]. However, if this were the case, it would be difficult to reconcile with the findings of

Sarlos et al., who reconstructed the UFB-binding complex in vitro using recombinant proteins that were mostly purified from *E. coli* and yeasts, where specific mitotic phosphorylation is unlikely to be present[8]. More importantly, as demonstrated here, inhibition of mitotic kinases and mutations of specific mitotic phosphorylation sites on BLM indeed stimulate, rather than compromise, the cellular activity of the UFB-binding complex (Fig. 7b, c). In line with this observation, we also show that the RIF1-PP1s activity is required to promote the assembly of the UFB-binding complex induced by PLK1 inhibition, presumably through counteracting the CDK1 inhibitory phosphorylation. However, RIF1 is not required for the recruitment of BTRR complex to PICH-bound UFBs during anaphase[9]. Given that the activity of mitotic kinases sharply decreases at the metaphase-anaphase transition, where multiple protein phosphatases are activated, it is plausible the inhibitory phosphorylation on the BTRR complex can be quickly removed by other activated phosphatases. This proposal is in line with our findings that the RIF1-PP1s inactivation do not completely abolish the activation of mitotic BTRR complex by PLK1 inhibition. In addition, we show that abolishing the mitotic phosphorylation sites surrounding the TOP3A/RMI1-binding interfaces of BLM can restore their interaction at centromeres. Therefore, our findings collectively support the model that mitotic phosphorylation acts negatively to regulate the UFB-binding factors. Notably, while CDK1 can attenuate stable interaction between BLM and the TRR subcomplex, the phosphorylation sites mapped across the TOP3A/RMI1 binding interfaces do not align with CDK1 motif consensus. This may suggest the involvement of another kinase(s), downstream of CDK1, in mediating the blocking of the protein-protein interactions. Nevertheless, the characterisation of the BLM and TOP3A interaction also reshapes our understanding on the assembly mechanism of the UFB-binding complex. Although it has been suggested that the BLM helicase and the TRR complex can independently bind to PICH on UFB molecules[8], our cellular analyses demonstrate that this rarely occurs in anaphase cells. This can be explained by the fact that the binding affinity of PICH for these individual factors is 3 to 5-fold lower than that of the whole BTRR dissolvasome in vitro[8]. Our data also aligns well with a previous study showing that the UFB localisation of both BLM and TOP3A proteins is severely compromised in RMI2-deficient cells[66]. Therefore, multiple PICH-interacting surfaces may be required to secure the stable formation of UFB-binding complexes. Interestingly, within the BTRR complex itself, its formation also requires stable interaction of BLM to both TOP3A and RMI1 subunits. Conceivably, these multiple low-affinity interactions, within the BTRR complex and to PICH, likely enable exquisite control of the UFB-binding complex assembly through phosphorylation of different interfaces. This aligns with our hypothesis that regulating the BTRR complex is one possible means of restraining the UFB-binding complex activity in mitosis.

Disabling CDK1 is sufficient to override the suppression of the BTRR complex assembly at K-chromatin, but that is inefficient to trigger centromere unwinding. As shown here, this is because of the PLK1-

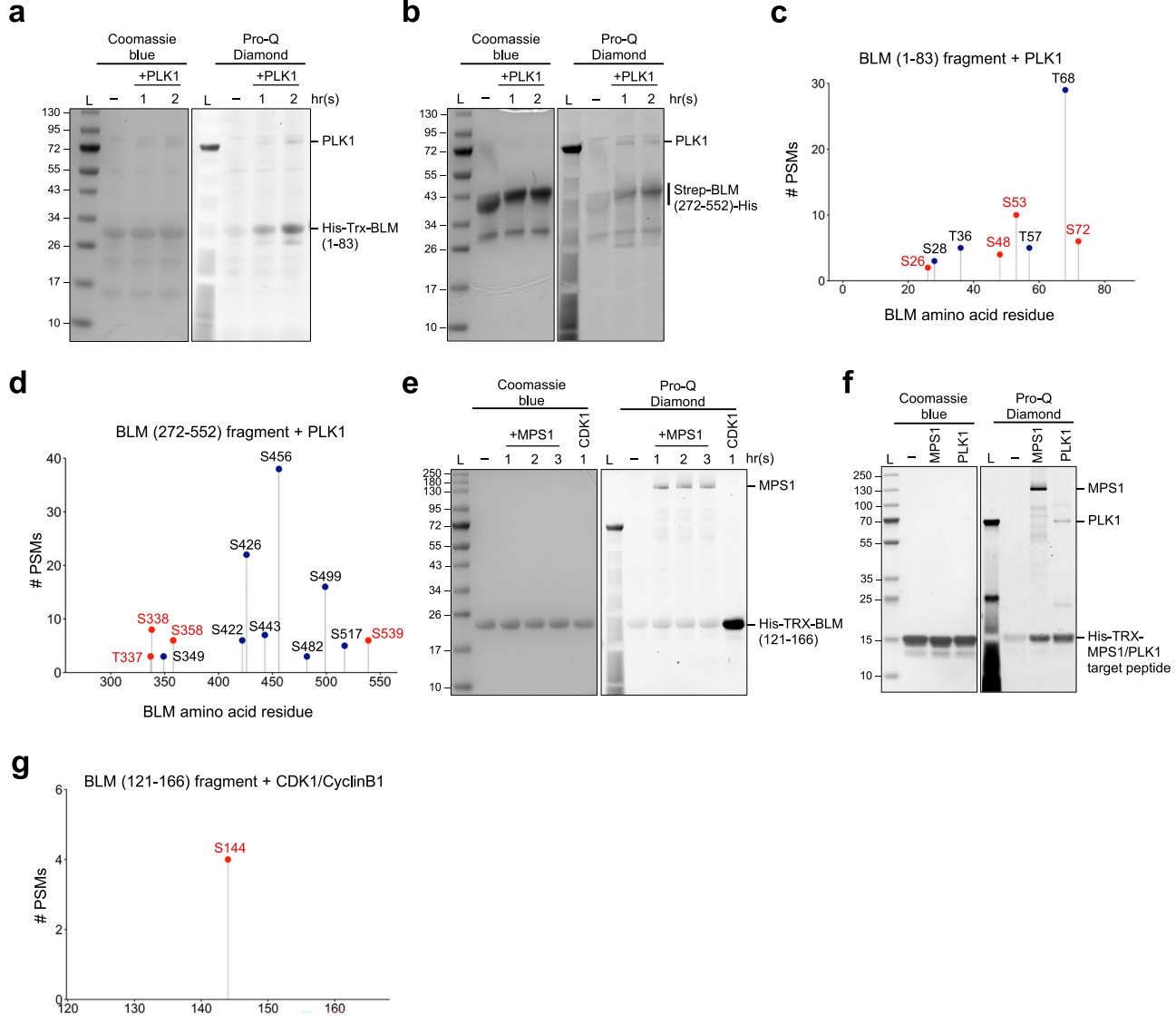

**Fig. 5 | CDK1 and PLK1 phosphorylate purified recombinant BLM fragments at specific sites in vitro. a** PLK1 phosphorylates recombinant BLM (1-83) fragments in vitro. Purified BLM fragments were incubated with PLK1 for the indicated times. Phosphorylation was assessed by Pro-Q Diamond staining before mass spectrometry (MS) phosphorylation analysis. **b** Same experimental steps as shown in **a** but using recombinant BLM (272-552) fragments. **c**, **d** Lollipop plots showing PLK1 in vitro phosphorylation sites within residues 1-83 and 272-552 of the BLM fragments by MS. BLM peptide coverages are 100%. PSM stands for peptide-spectrum match. Same phosphorylation sites were found from samples of both timepoints. Red-colour residues are the putative PLK1 target sites, whereas the black ones were observed in mitotic cells in previous phospho-proteomic studies. **e** CDK1/CyclinB1, but not MPS1, phosphorylates BLM (121-166) fragments in vitro. Phosphorylation was assessed by Pro-Q Diamond gel staining before MS. **f** Purified positive control peptides containing MPS1/PLK1 targeting sequence were actively phosphorylated by recombinant MPS1 and PLK1 kinases after 1 h incubation. **g** CDK1/CyclinB1 phosphorylates serine 144 on BLM in vitro. All experiments of different timepoints were performed in technical duplicates. The same gels were sequentially stained by Pro-Diamond and Coomassie blue dyes. Coomassie blue staining served as a loading control. L, protein ladder. Source data are provided as a Source data file.

mediated hyperphosphorylation on BLM (Fig. 7c). However, we do not think that the phosphorylation directly inhibits BLM enzymatic activity because phosphorylated BLM isolated from metaphase-arrested cells has been shown to be proficient in DNA unwinding in vitro[21]. Intuitively, one might expect a direct allosteric regulation of the helicase domain to be more efficient to suppress BLM, such as altering its ability to bind ATP. However, the inhibitory phosphorylation sites identified here reside across the disordered N-terminal region rather than on the core helicase domain. This may be attributed to higher accessibility of the mitotic kinases to the less structured part of the protein, but how does phosphorylation at the N-terminus of BLM interfere DNA unwinding on centromere chromatin? One possibility may be through hindering BLM's accessibility to the DNA substrates and/or co-activators. It is notable that a cluster of the inhibitory phosphorylation sites is located between the putative TOPBP1 binding region, close to the essential serine 304 residue for the TOPBP1-BRCT5 domain interaction[29], and the double helical bundle in the N-terminal (DHBN) domain responsible for BLM dimerization[28,49]. Mutations in either Ser304 or the DHBN domain have been shown to compromise BLM anti-crossover but not DNA unwinding activity[18,28]. Thus, it is conceivable that the cluster of phosphorylation might interfere with these cofactors, thereby limiting BLM activities at centromeres. Further structural analyses will be required to address these possibilities. In conclusion, we propose that the Bloom syndrome helicase complex can act as a double-edged sword for genome stability. While the active complexes are essential in suppressing genome instability, the lack of temporal regulation can seriously

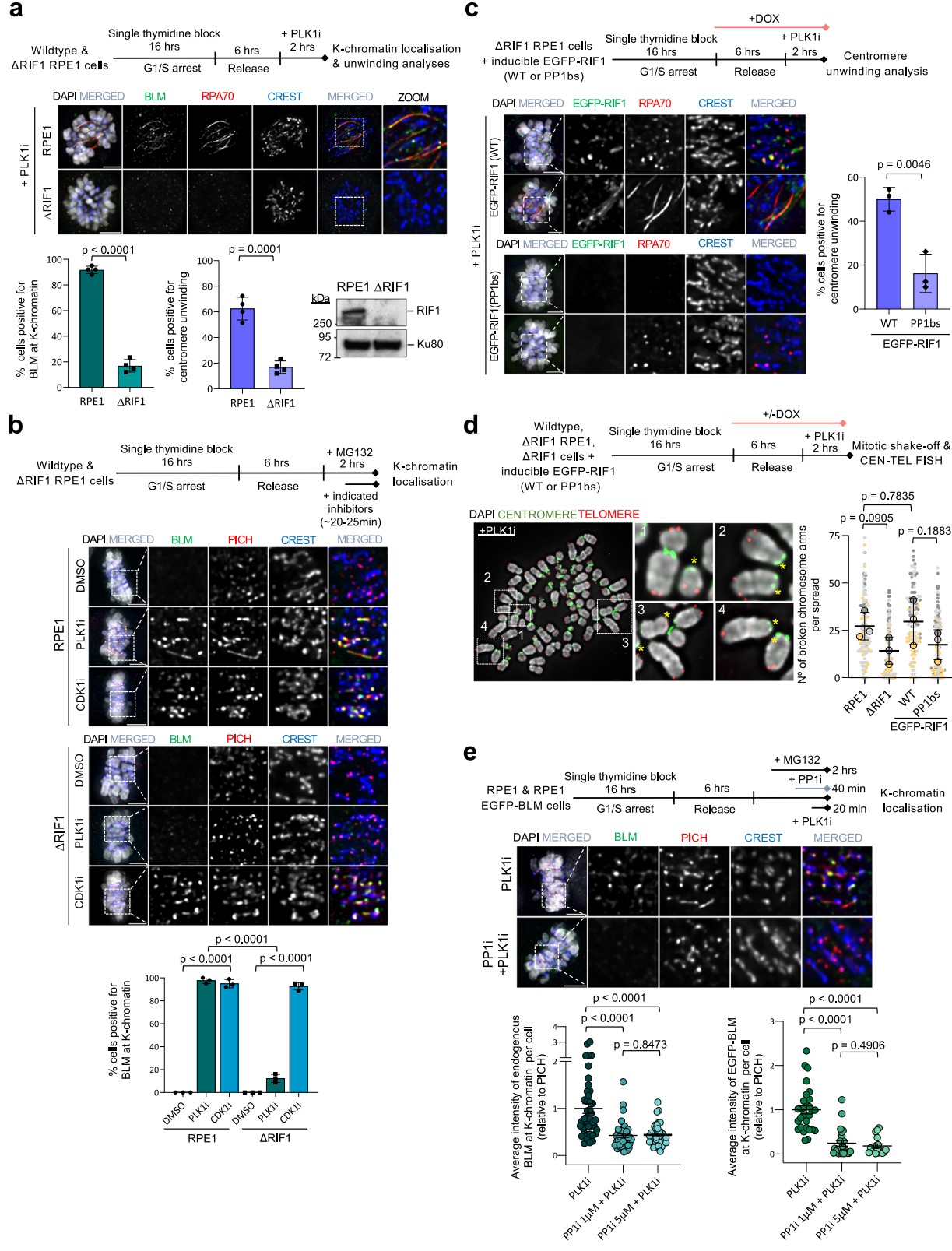

endanger chromosome integrity, particularly at the regions of centromere chromatin.

## Methods

### Cell culture and cell lines

Cell lines HeLa, U2OS, HCT116, and HEK293T and their derivates were cultured in DMEM (Sigma) supplemented with 10% foetal calf serum (FCS) (Gibco), 1% L-glutamine, and 1% Penicillin/Streptomycin antibiotics. RPE1-hTERT and its derivates were cultured in DMEM/F-12 medium (Sigma) supplemented with 10% FCS (Gibco) and 1% Penicillin/Streptomycin antibiotics. HAP1 and its derivates are cultured in IMDM (Gibco) supplemented with 10% FCS (Gibco) and 1% Penicillin/ Streptomycin antibiotics. All human parental cell lines used were obtained from the Cell Bank at the Genome Damage and Stability

**Fig. 6 | RIF1 and PP1s facilitate the activation of UFB-binding complex.**
**a** Experimental setup (top). Representative images and percentages of mitotic cells positive for BLM at K-chromatin and centromere unwinding are shown. Numbers of cells analysed from four independent experiments: RPE1 $n = 73, 72, 68, 69$; ΔRIF1 $n = 70, 74, 74, 77$; mean ± S.D. is shown. Western blotting shows RIF1 levels. **b** K-chromatin localisation of BLM in wildtype and ΔRIF1 RPE1 cells. Experimental setup (top). Representative images and percentages of metaphase-arrested cells positive for BLM at K-chromatin are shown. Numbers of cells analysed from three independent experiments: RPE1 DMSO $n = 62, 58, 64$; PLK1i $n = 60, 60, 85$; CDK1i $n = 50, 50, 98$; ΔRIF1: DMSO $n = 54, 58, 63$; PLK1i $n = 50, 77, 75$; CDK1i $n = 50, 60, 100$; mean ± S.D. is shown. **c** Centromere unwinding in ΔRIF1 RPE1 cells with wildtype (WT) or PP1s-binding mutant (PP1bs) of EGFP-RIF1. Representative images and percentages of mitotic cells positive for centromere unwinding are shown. Numbers of cells analysed from three independent experiments: WT $n = 70, 59, 88$ and in PP1bs $n = 70, 80, 84$; mean ± S.D. is shown. **d** Centromere disintegration analysis.

Representative images of metaphase chromosomes after PLK1 inhibition. Centromeres and telomeres are labelled by FISH PNA probes. Asterisks indicate chromatin with centromere breakage. Quantification of broken whole chromosome arms per metaphase spread. Numbers of metaphase spreads analysed from three independent experiments: RPE1 $n = 31, 39, 73$; ΔRIF1, $n = 39, 33, 35$; ΔRIF1 + EGFP-RIF1(WT) $n = 37, 33, 41$; ΔRIF1 EGFP-RIF1(PP1bs) $n = 42, 39, 34$. mean ± S.D. is shown. **e** Experimental outline of co-treatments of PP1s and PLK1 inhibitors. Representative images of BLM K-chromatin localisation in RPE1 metaphase cells. Graphs show quantification of average centromeric intensity of BLM relative to PICH per cell in RPE1 (left) and RPE1 EGFP-BLM (right) cells. Data is normalised to the average intensity in PLK1i condition. Numbers of cells analysed in each condition, RPE1: PLK1i $n = 43$; PLK1i + 1 μM PP1i $n = 41$; PLK1i + 5μM PP1i $n = 40$; RPE1 EGFP-BLM: PLK1i $n = 29$; PLK1i + 1μM PP1i $n = 24$; PLK1i + 5μM PP1i $n = 17$; mean ± S.E.M is shown. Scale bars, 5 μm. All $p$ values are calculated by unpaired two-tailed $t$-test. Source data are provided as a Source data file.

Centre, which were verified by ATCC's cell line authentication service. All cell lines were periodically tested and confirmed free from mycoplasma (Lonza Mycoplasma testing kit). All cell lines were maintained at 37 °C in a humidified atmosphere containing 5% $CO_2$. Stable cell lines in this study were generated by stably transfecting plasmids containing gene of interests either by electroporation using Neon Transfection Kit (ThermoFisher; MPK1025) (RPE1 hTERT cells) at 1350 V, for 20 ms with 2 pulses, or lipid-mediated delivery using FuGENE HD (Promega; TM328) according to the manufacturer's guidelines. FACS cell sorter (BD FACS Melody) was used to sort cells expressing EGFP or mCherry tagged proteins to generate stable cell lines. The sorted cells were subjected to antibiotic selection: 800 mg/ml–1200 mg/ml of G418 for 10 days; 0.25 μg/ml–0.5 μg/ml of Puromycin for 3 days. Mutations were introduced into plasmids of interest using QuikChange XL Site-Directed Mutagenesis Kit (Agilent Technologies; 200516) according to the manufacturer's protocol.

## Chemicals and small molecule inhibitors
Doxycycline (Merck; D5207, 0.25–1 μg/ml) was used to induce expression gene of interest using Sleeping Beauty (SB) transposon system (pSBtet vector)[67]. Thymidine (Sigma; T9250, 2 mM), nocodazole (Sigma; SML1665, 50–100 ng/ml), MG132 (Sigma; 474790, 20 μM) to control the cell cycle and arrest the cells. ICRF193 (Merck; I4659, 0.1–1 μM) was used to induce UFBs. Mitotic kinase inhibitors: AZD1152 (Sigma-Aldrich; S1147, 100 nM), BI2536 (Cayman Chemical; 17385, 100 nM), MLN8237 (Selleck Chemicals; 1133, 50 nM), Reversine (Axon MedChem; 1629, 1 μM), RO-3306 (Sigma-Aldrich; SML0569, 7.5–9 μM), and Protein Phosphatase 1 inhibitor: Tautomycetin (Tocris Bioscience; 2305, 1 or 5 μM).

## RNA interference
Cells were transfected with siRNA oligonucleotides using Lipofectamine RNAiMAX transfection reagent (Thermo Fisher Scientific; 13778075) following the manufacturer's guidelines.

Non-targeting siRNA pool (Dharmacon ON-TARGET plus Non-targeting Pool–D-001810-10-05. UGGUUUACAUGUCGACUAA; UGGUUUACAUGUUGUGUGA; UGGUUUACAUGUUUUCUGA; UGGUUUACAUGUUUUCCUA).

BLM siRNA sequence (Dharmacon ON-TARGET plus Individual–J-007287-08-0005. GGAUGACUCAGAAUGGUUA).

TOP3A siRNA sequence (Ambion Thermo Fischer 4392420. CGGCUUGCCUAGUUCUCUA).

SGO1 siRNA sequence (Dharmacon ON-TARGET plus SMARTpool –L-015475-00-0005. CAGCCAGCGUGAACUAUAA; GUUACUAUCUCACAUGUCA; AAACGCAGGUCUUUUAUAG; GUGAAGGAUUUACCGCAAA).

## Fluorescence-activated cell sorting (FACS)
Cells were trypsinised, washed with PBS, and fixed with 70% ice-cold ethanol, adding dropwise whilst using a vortex. Cells were

resuspended in Propidium Iodide (PI) /RNase staining buffer (9.5 ml 1× PBS, 400 μl 1 mg/ml PI solution, 10 mg/ml RNaseA). The resuspended cells were passed through a Falcon cell strainer into a round-bottom tube (Falcon Corning; 352235) for flow cytometry analysis. Cell cycle profiles were then determined and analysed using BD Accuri C6 sampler.

## Immunofluorescence staining
For immunostaining analysis, cells were seeded onto cover glass of No. 1.5 or 1.5H and treated as indicated. Cells were fixed with Triton X-100-PFA buffer (250 mM HEPES pH 7.4, 1× PBS, 0.1% Triton X-100, 4% methanol-free paraformaldehyde) at 4 °C for 20 min, or with PBS–PFA buffer (1× PBS, 3.7 % methanol-free paraformaldehyde) at room temperature for 10 min. Fixed cells washed with PBS and permeabilised with 0.1% of Triton X100 and 0.5 % FCS in PBS for 20 min at 4 °C, followed by blocking (1× PBS, 0.5 % FCS) at room temperature for 10 min. Cells were incubated with primary antibody at 37 °C for 90 min followed by secondary antibody incubation at room temperature for 30 min. Slides were washed with 1× PBS for 5 times at room temperature after antibody incubation. Cells were washed with ultra-pure water, and coverslips were air dried before mounting using Vectashield mounting medium, either with or without DAPI. The latter is used if nuclei were pre-stained with Hoechst H33342 (Invitrogen; C10637-G) after the secondary incubation.

Fluorescent images were acquired in a Zeiss AxioObserver Z1 epifluorescence microscopy system equipped with 40x/1.3 oil Plan-Apochromat, 63x/1.4 oil Plan-Apochromat, and 100x/1.4 oil Plan-Apochromat objectives and a Hamamatsu ORCA-Flash4.0 LT Plus camera. Z-stack images were acquired at 0.2 μm intervals covering a range from 2 to 8 μm by using ZEN blue software. Image deconvolution was performed using Huygens Professional deconvolution software with a measured point-spread-function (PSF) generated by 200 nm-diameter TetraSpeck microspheres (ThermoFisher; T7280). Classical maximum likelihood estimation method with iterations of 40–60 and a range of 20–60 signal-to-noise was applied. Fiji was used to generate the representative images.

Primary antibody dilution: rabbit Aurora B (Abcam; ab45145, 1:100), rabbit BUBR1 (Abcam; ab209998, 1:100), goat anti-BLM (Santa Cruz; sc-7790, 1:100), mouse anti-BLM (Santa Cruz; sc-365753, 1:100), rabbit anti-BLM (Abcam; ab2179, 1:100), mouse anti-CENP-A (Abcam; ab13939, 1:100), human anti-CREST (Immuno Vision; HCT-0100, 1:400), mouse Cyclin B1 (BD Biosciences; 610219, 1:100), alpaca anti-GFP-Atto488 (Chromotek; gba488, 1:200), alpaca anti-GFP-AF488 (Chromotek; gb2AF488, 1:200), rabbit anti-NUF2 (Abcam; ab122962, 1:200), mouse anti-PICH (Abnova; H00054821-B01P, 1:100), rabbit anti-PICH (Abnova; H00054821-D01P 1:100), rabbit anti-RIF1 (Bethyl Lab; A300-568A, 1:100), rabbit anti-RIF1 (Bethyl Lab; A300-569A, 1:100), rabbit anti-RPA70 (Abcam; ab79398, 1:200) and rabbit anti-TOP3A (Abcam; ab108493, 1:100); Secondary antibody dilution:

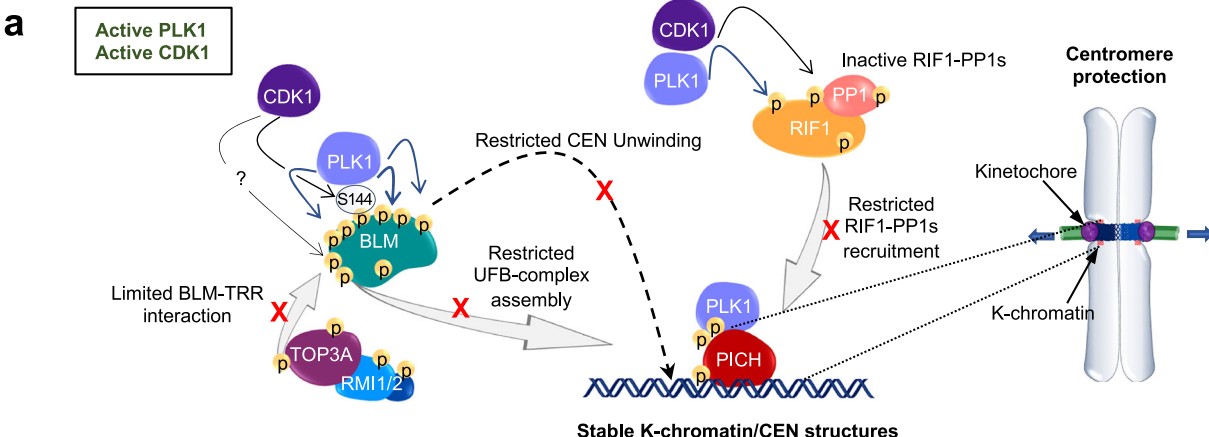

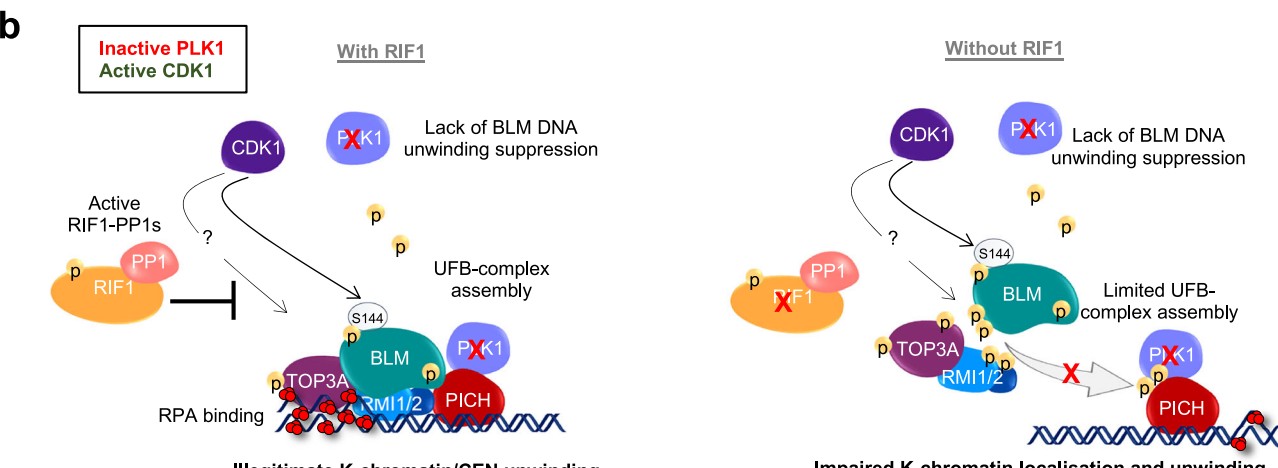

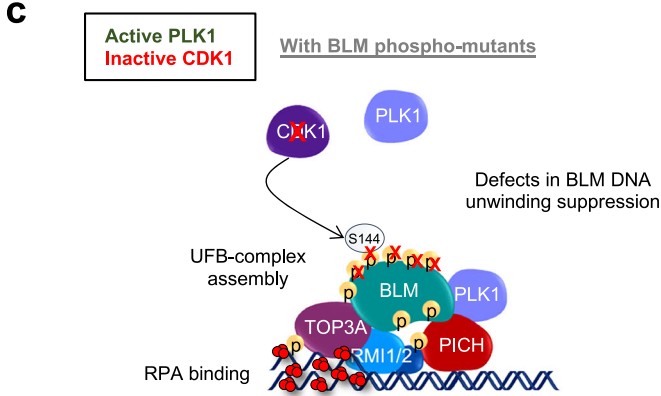

**Fig. 7 | A model of the centromere protection pathway. a** During chromosome alignment, CDK1 and PLK1 cooperate to promote multiple phosphorylation on different subunits of the ultrafine DNA bridge (UFB)-binding complexes, including BLM helicase and PICH DNA translocase. The inhibitory phosphorylation destabilises the assembly of the BTRR complex, RIF1-PP1s recruitment, limiting their association with PICH at kinetochore-associated (K)-chromatin in centromeres. The PLK1-mediated hyperphosphorylation on BLM also suppresses illegitimate DNA unwinding on centromeric DNA. Therefore, mitotic cells can maintain robust centromere structures to counteract bipolar spindle pulling forces and establish proper chromosome biorientation. **b** In the absence of PLK1 activity, the inhibition of BTRR/UFB-binding complexes is relieved in a RIF1-PP1s-dependent manner, resulting in illegitimate assembly of the BTRR/UFB-binding complexes at K-chromatin. The unleashing of BLM activity after the loss of PLK1-mediated phosphorylation leads to centromeric DNA unwinding as visualised by RPA binding and results in centromere disintegration (left). In the absence of RIF1-PP1s, this prevents efficient removal of the inhibitory phosphorylation by CDK1, limiting the initial assembly the BTRR/UFB-binding complexes at K-chromatin (right). **c** BLM phosphomutants lacking either serine 144 or the PLK1-mediated phosphorylation sites become dominantly active in centromeric DNA unwinding once the complexes assembly at K-chromatin is permitted after CDK1 inhibition.

donkey anti-goat AF488 (Invitrogen; 1463163, 1:500), donkey anti-mouse AF488 (Invitrogen; A-21202, 1:500), donkey anti-rabbit AF488 (Invitrogen; A-31570, 1:500), donkey anti-mouse AF555 (Invitrogen; A-31570, 1:500), donkey anti-rabbit AF555 (Invitrogen; A-31572, 1:500), donkey anti-mouse AF647 (Invitrogen; A-31571, 1:500), donkey anti-rabbit AF647 (Invitrogen; A-31573, 1:500) and goat anti-human AF650 (Abcam; ab98622, 1:500).

## Time-lapse live-cell microscopy

For live-cell tracking, cells were seeded on 2-well or 4-well tissue culture chambers, cover glass II (Sarstedt), and were monitored using a Zeiss AxioObserver Z1 epifluorescence microscopy system equipped with a heating and $CO_2$ chamber (Digital Pixel). The conditions were kept at 37 °C and 5 % $CO_2$. Image acquisition was performed using 40×/0.95 Korr Plan-Apochromat (correction collar adjusted to 0.17) or 40×/1.3NA oil Plan-Apochromat objectives and a Hamamatsu ORCA-Flash4.0 LT Plus camera. ZEN Blue software was used for image and movie acquisition with 3 to 6 z-stacks of 1 µm z-intervals. Images were processed using ImageJ/Fiji software.

## High-precision deconvolution microscopy

The microscope objectives and system were regularly aligned using 200 nm-diameter TetraSpeck microspheres (ThermoFisher) to ensure the minimal interference of chromatic aberration. Chromatic shifts, if any, were corrected by ZEN Blue software. This allows precise determination of the relative positions between kinetochore proteins and PICH labelled by different fluorophores. All images were subject to deconvolution by using Huygens Professional deconvolution software as described above.

## Stimulated emission depletion (STED) super-resolution nanoscopy

STED images were acquired using the Expert Line Easy3D STED microscope system (Abberior Instruments GmbH). The setup was based on an inverted Olympus IX83 microscope equipped with an objective 100×1.4NA Oil UPLSAPO100XO (Olympus), an automatic XYZ stage, and a real-time autofocusing device. Multichannel confocal images were obtained using excitation lasers of 405 nm, 488 nm, 561 nm, and 640 nm. Stimulated depletion was achieved with a 775 nm STED laser (Abberior Instruments GmbH). Fluorescence signals in predefined channels were detected using avalanche photodiode (APD) detectors. Red and far-red channels STED super-resolution imaging of fixed samples were obtained with typical ~60 nm XY spatial resolution. Pixel sizes for STED and confocal imaging are 10–15 nm and 100 nm, respectively. The imaging acquisition process was controlled by the Imspector Software (Abberior Instruments Development Team, Imspector Image Acquisition and Analysis Software). H33342 (Invitrogen; C10637 G, 0.25 µg/ml) was used to stain the DNA prior fixation. All primary antibodies were used in a dilution of 1:100. Secondary antibody: goat anti-mouse STAR ORANGE (Abberior; ST580-1001, 1:200), goat anti-rabbit STAR ORANGE (Abberior; ST580-1002, 1:200), goat anti-mouse STAR RED (Abberior, STRED-1001, 1:200), and goat anti-rabbit STAR RED (Abberior, STRED-1002, 1:200). Following immunofluorescence procedure, cells were washed with ultra-pure water, and coverslips were air dried before mounting using Mount Liquid Antifade (Abberior). The excitation and depletion laser powers for STAR RED are 3–5% and 25–45%, respectively. The excitation and depletion laser powers for STAR ORANGE are 15–25 and 25–35%, respectively. Line accumulation is set between 15 and 20. Images were further processed using ImageJ/Fiji software.

## Imaging analysis

CENPA-PICH/CENPA-NUF2 coordinates: The measurements of the distance between foci in two channels of an image were done by using Spot Pair Distance tool in Fiji software. The tool searches within a focus/box radius, typically± 5px, for a local maximum in the two pre-selected analysis channels. The centre-of-mass around each maximum, typically± 2px, is computed as the centre of intensity for each channel. Dragging from the clicked point creates a reference direction. The Euclidean distance between the centres is reported, optionally with the signed XY distance and angle relative to the reference direction. Visual guides are overlaid on the image to assist in spot selection and direction orientation. Available in the latest GDSC (Genome Damage and Stability Centre) Fiji plugins.

UFB intensity measurements: UFBs were stained for the UFB-binding complex components, including BLM, TOP3A, and PICH, following immunofluorescence protocol. Fiji software was used to determine the intensity of individual UFBs by drawing a scanline along the entire length of each UFB thread using PICH channel as a reference, which allowed the measurement of the absolute intensity of other channels corresponding to either BLM or TOP3A. The intensity of different channels was subjected to background correction prior to further analysis.

Centromere localisation analysis: The intensity of each centromere/cluster was determined using the Find Foci GUI tool in Fiji software, which locates all the points of maximum intensity in the centromeric regions. This was used to generate a mask image containing all the pixels in the peak region above background in the CENPB-mCherry or PICH channel by using the Otsu thresholding method. The intensities of different channels within the centromeric masked regions were then measured using the Mask Analyser Channels plugin available in the latest GDSC (Genome Damage and Stability Centre) Fiji plugins.

## Western blot

Cells were trypsinised and washed with ice-cold PBS once followed by protein extraction in Lysis Buffer A or B (Lysis buffer A: 50 mM Tris-HCl pH7.5, 300 mM NaCl, 5 mM EDTA, 1% Triton X-100, 1 mM DTT, 1 mM PMSF; Lysis buffer B: 50 mM Tris-HCl pH7.5, 150 mM NaCl, 1% Triton X-100, 1.25 mM DTT and 1 mM PMSF, 25U/ml benzonase (Sigma-Aldrich; E1014). The lysis buffers were supplemented with Protease inhibitor cocktail Complete protease inhibitor (Roche; 11697498001) 1 tablet/50 ml and, when required, phosphatase inhibitor PhosSTOP (Roche; 4906845001) 1 tablet/10 ml. Cell lysate was incubated on ice for 15–20 min and centrifuged at 15,000 rpm for 20 min at 4 °C. Protein concentration was quantified using a Bradford assay (Bio-Rad). Samples were separated by SDS-PAGE, transfer onto Amersham Hybond PVDF membranes, 0.2 µm (GE Healthcare Lifesciences, 10600021), and blotted with the indicated antibodies following standard procedures.

Primary antibodies: mouse anti-β-actin (Sigma; A5441, 1:5000), goat anti-BLM (Bethyl Lab; A300-120, 1:2000), rabbit anti-BLM (Abcam; ab2170, 1:500), rabbit anti-GFP (Abcam; ab290, 1:2000), rat anti-GFP (Chromotek; 3h9-100, 1:2000), rabbit anti-Ku80 (Abcam; ab80592, 1:5000), mouse anti-PICH (Abnova; H00054821-BO1P, 1:500), rabbit anti-RIF1 (Bethyl Lab; A300-568A, 1:1000), rabbit anti-RIF1 (Bethyl Lab; A300-569A, 1:1000), rabbit anti-TOP3A (Proteintech; 14525, 1:500), mouse anti-SGO1 (Abcam; ab58023, 1:500). HRP-conjugated secondary antibodies: rabbit anti-goat (Agilent; P0160, 1:60000), goat anti-mouse (Abcam; ab6780, 1:25000), donkey anti-rabbit (ECL/Sigma; NA9340, 1:40000), and goat anti-rat (ECL/Sigma; GENA935, 1:25000).

## Co-immunoprecipitation

HEK293T cells were transiently transfected with EGFP-BLM or EGFP-TOP3A plasmids, or ΔBLM HAP1 cells with EGFP-TOP3A plasmids. Additionally, HeLa cells stably expressing EGFP-TOP3A were also used to perform mitotic analysis of protein-protein interaction. The cell pellet was lysed in the lysis buffer A or B as the immunoblot procedure. Input samples were collected prior to standard immunoprecipitation procedure. The lysate was applied on the equilibrated GFP-

Trap_Magnetic Agarose beads (GFP-Trap_MA) or GFP-Trap_Dynabeads (GFP-Trap_M-270) and rotated for 1 h at 4 °C. Then the samples were collected by a magnet and washed three to five times in the diluting buffer (50 mM Tris-HCl pH7.5, 150 mM NaCl, 0.5 mM EDTA). Concentrations of NaCl varied, ranging from 150 mM to 450 mM when indicated. The proteins bound to the beads were resuspended in 1X Laemmli Sample Buffer (BioRad) with 5% β-mercaptoethanol (Sigma, M6250) and boiled for 5 min. Supernatant was analysed by Western blot.

Endogenous BLM was pull down in HeLa, U2OS, or RPE1 PICH-mAID-mClover3 cells using rabbit anti-BLM (Abcam; ab2179). Endogenously mClover3-tagged PICH proteins were pull down from RPE1 PICH-mAID-mClover3 cells. Cells were lysed in lysis buffer B containing benzonase. Input samples were collected prior to standard immunoprecipitation procedure. Following a pre-incubation step of the cell lysates with the antibody for 1 h at 4 °C, the mixture was applied to DynaGreen™ Protein A/G magnetic beads (Invitrogen; 80104 G) and rotated for 1 h at 4 °C. For PICH-mAID-Clover3 protein pull down, GFP-Trap_Dynabeads (GFP-Trap_M-270) were used. Then the samples were washed three times in the diluting buffer (50 mM Tris-HCl pH 7.5, 150 mM NaCl, 0.5 mM EDTA). Various concentrations of NaCl were applied during the washing step as indicated in the experiments testing BLM and TOP3A interaction. The proteins bound to the beads were resuspended in 1X Laemmli Sample Buffer (BioRad) with 5% β-mercaptoethanol (Sigma; M6250) and boiled for 5 min. Supernatant was analysed by Western blot.

## Mass spectrometry (MS)

HAP1 ΔBLM stably expressing EGFP-BLM wildtype was used for mass spectrometry analysis following immunoprecipitation procedures using GFP-Trap assay. The cell pellet was lysed in lysis buffer B (see Western Blot procedure). The lysate was applied on the equilibrated GFP-Trap_Magnetic Agarose beads (GFP-Trap_MA) and rotated for 1 h at 4 °C. Then the samples were collected by a magnet and washed three to five times in the diluting buffer (50 mM Tris HCl pH 7.5, 150 mM NaCl, 0.5 mM EDTA). The proteins bound to the beads were resuspended in 1X Laemmli Sample Buffer (BioRad) with 5% β-mercaptoethanol (Sigma; M6250) and boiled for 5 min. Supernatant was run in an SDS-PAGE gel, which was stained by Coomassie blue. Bands corresponding to the EGFP-BLM size were extracted and frozen at −80 °C. Liquid Chromatography with tandem mass spectrometry (LC-MS) experiment was performed by the BSRC Mass Spectrometry and Proteomics Facility at the University of St. Andrews. In brief, gel bands were reduced, alkylated, and digest followed phospho-peptide enrichment on TiO2 (Titanium dioxide) columns. The eluted phospho-peptides were subsequently analysed on their LC-ESI-MS instrument. The data was searched against the NCBI (National Centre for Biotechnology Information) database containing all species and analysed using the Mascot Server. In a second run of mass spectrometry the immunoprecipitation of EGFP-BLM wildtype was carried out in a similar manner. A fraction of the beads was resuspended in 1X Laemmli Sample Buffer (BioRad) with 5% β-mercaptoethanol (Sigma, M6250) and boiled for 5 min. This was analysed by silver staining, using Silver Staining Kit (Pierce) and Western Blot. After the washing steps, beads were additionally washed three times with 1× PBS, to remove detergents and free amines, and frozen at −80 °C. Trypsin digestion and TMT-3plex (Tandem mass tag) mass spectrometry were performed at Institute of Cancer Research (ICR). In brief, samples were solubilised, reduced, alkylated, and digested followed by labelling with the TMTpro multiplexing reagents (Thermo Scientific). Prior to LC-MS analysis of TMT-labelled peptides were fractionated with high-pH reversed-phase (RP) chromatography and enriched with immobilised metal ion affinity chromatography. The SequestHT search engine was used to analyse the data.

## In vitro kinase assays and MS Phosphorylation analysis

Protein Purification of BLM Fragments: Purification of recombinant proteins was done using *E.coli* bacterial expression system. BL21(DE3) cells carrying a plasmid encoding HIS-TRX-(3 C)-BLM-(1-83), HIS-TRX-(3 C)-BLM-(121-166), Strep-(3 C)-BLM-(272-552)-HIS or HIS-TRX-(3 C)-MPS1/PLK1 control peptide (GGSGLLLDSTLSINWGGS)[50], were grown in TurboBroth at 37 °C in an orbital-shaker incubator until an $OD_{600}$ of 0.6–0.8 was reached and induction of recombinant protein expression was performed by the addition of 0.2 mM IPTG (isopropyl β-d-1-thiogalactopyranoside) (Generon Ltd, Slough, UK). Cultures were then incubated at a lower temperature of 20 °C in an orbital-shaker incubator for a period of ~16 h, before harvesting of cells by centrifugation at 4000 × g for 20 min. Cell pellets were resuspended in lysis buffer containing 25 mM HEPES pH 7.5, 200 mM NaCl, 0.5 mM TCEP, and 10 mM imidazole supplemented with cOmplete EDTA free protease tablets (Roche), disrupted by homogenisation and sonication, and the resulting lysate clarified by centrifugation at 40,000 × g for 60 min at 4 °C. The supernatant was applied to a 1 ml TALON column, then washed with lysis buffer supplemented with 10 mM imidazole, before retained protein was eluted using lysis buffer supplemented with 300 mM imidazole.

In Vitro Kinase Assays: GST-tagged MPS1/TTK kinase (ab89589) and His-tagged PLK1 kinase (ab51426) were purchased from Abcam. His/Strep-tagged CDK1/Cyclin B complex was produced as previously described in refs. 68,69. Purified recombinant proteins BLM 1-83, BLM 121-166, BLM 272-552, and MPS1/PLK1 control peptides were treated with the corresponding kinases as indicated. To exchange the buffer for optimal conditions for the reaction, the protein solution was subjected to buffer exchange using a spin-column (Spin-Trap G25, GE Healthcare, 28922527) according to company protocol. Buffer composition: 20 mM HEPES NaOH pH 7.5, 150 mM NaCl, 0,5 mM TCEP. The kinase reaction was performed in a total reaction volume of 60 μl consisting of 50 μM pf protein substrates and 0.3–1 μM of kinases in reaction buffer (20 mM HEPES NaOH pH 7.5, 150 mM NaCl, 0,5 mM TCEP, 2.4 mM ATP) at 30 °C for 1–3 h as indicated. The reaction products were analysed by SDS-PAGE, stained with Pro-Q™ Diamond Phosphoprotein Gel Stain (Invitrogen, P33300) according to manufacturer's instructions and/or mass spectrometry.

Sample preparation for mass spectrometry: Following the in vitro kinase reaction, protein samples (BLM fragments incubated with or without different kinases) were reduced with 5 mM TCEP at 37 °C for 1.5 h and alkylated with 10 mM iodoacetamide at room temperature for 1 h. Samples were then purified using PD Spintrap™ G-25 (Cytiva) and digested with trypsin (Promega) overnight at 37 °C in 50 mM ammonium bicarbonate buffer. Peptides were desalted with C18 spin columns (Pierce) and dried via vacuum centrifugation. Peptide samples were stored at −80 °Cuntil further analysis.

LC-MS/MS analysis: Samples were analysed by LC-MS/MS using an OrbiTrap Exploris 480 mass spectrometer (Thermo). Samples were injected onto an Easy Spray PepMap C18 column (75 μm id × 50 cm, 2 μm particle size; Thermo) and separated over a 60-minute method. The gradient for separation consisted of 1–35% mobile phase B at a 350 nl/min flow rate, where mobile phase A was 0.1% formic acid in water, and mobile phase B consisted of 0.1% formic acid in 80% ACN. The OrbiTrap Exploris 480 was operated in data-dependent mode using FAIMS using 2 compensation voltages −45V and −65V for 2 and 1 s, respectively, during which maximal number of precursors were selected for subsequent fragmentation. Resolution for the precursor scan (m/z 375–1200) was set to 60,000, while MS/MS scans resolution was set to 15,000. The normalised collision energy was set to 30% for HCD. Precursors with unknown charge or a charge state of 1 and ≥7 were excluded.

Data analysis: Raw data files were processed using Proteome Discoverer version 3.0.0.757 (Thermo Scientific). Using Sequest, peak lists were searched against a custom BLM fragment protein sequence

and kinase sequences acquired from Uniprot human database (downloaded in January 2025), as well as common contaminants database. The following parameters were used to identify tryptic peptides for protein identification: 10 ppm precursor ion mass tolerance; 0.02 Da product ion mass tolerance; up to two missed trypsin cleavage sites; (C) carbamidomethylation was set as a fixed modification; (M) oxidation and (S/T/Y) phosphorylation were set as variable modifications. The ptmRS node was used to identify the phosphorylation sites. Peptide data used in the BLM phosphorylation lollipop plot were first filtered to remove peptides with only a single PSM and those that had less than a 2-fold change average abundance across all samples, compared to the controls.

### AlphaFold2 protein structure prediction

For modelling of the BTRR complex, FASTA protein sequences were submitted to AlphaFold version 2.3.2[30] run on Apocrita, the Queen Mary, University of London High Performance Cluster. For both complexes, five models with five seeds were produced using multimer mode, and the top-ranked model was relaxed and taken for figure production using PYMOL Version 2.2.2 (The PyMOL Molecular Graphics System, Version 2.2.2 Schrödinger, LLC) to produce structural figures and AlphaPickle Version 1.5.4 (M. J. Arnold, 2021) for statistical plots.

### Metaphase spreads preparation

Cells were first synchronised using a single thymidine block and released to S phase for the indicated period. Two hours before the metaphase spread preparation, the cells were treated with 60 nM of the PLK1 inhibitor, BI2536 (Cayman Chemical; 17,385). Mitotic cells were collected by mitotic shake-off. Cell pellets were resuspended and mixed very gently in 10 ml of pre-warmed hypertonic solution (0.075 M KCl), followed by an incubation at 37 °C for 10 min. The swollen cells were centrifuged, fixed, and washed twice with Carnoy's Fixative (3:1; Methanol: Acetic Acid) according to the standard chromosome spread preparation. Chromosome spreads were dropped onto glass slides and stored at room temperature prior to FISH hybridisation.

### Centromere and telomere fluorescence in situ hybridisation (FISH)

The Peptide nucleic acid (PNA) probe hybridisation of chromosome spreads followed the manufacturer's guidelines (DAKO Agilent & PNAbio). Chromosome spreads were fixed in 3.7% PFA and washed using a gradient ice-cold ethanol wash series 70%, 90% and 100% EtOH. Slides were then air-dried prior to PNA probe addition to the spreads: centromere, CENTB-FAM (PNAbio; F3006), and Telomere TelG-Cy3 (PNA FISH kit, Dako Agilent; K532611-8 or PNAbio; F1006). A coverslip (18x18mm) was added to the probe and the slide was co-denatured at 80 °C for 30 sec to 1 min and hybridised at room temperature in a humidifying chamber for 2 h. Slides were then washed in Wash Solution (Dako Agilent) at 65 °C and dehydrated again in the ice-cold ethanol series. Slides were air dried and counterstained using Vectashield with DAPI.

### Sister chromatid exchange (SCE) assays

Cells were arrested in G1 using CDK4/6 inhibitor (Palbociclib, Merck; PZ0383, 1 μM) and released into S phase in the presence of EdU (Invitrogen; C10637, 0.5 μM) for 18–22 h. EdU was washed from the cells, which were kept in fresh medium for 12–18 h. To trap the mitotic cells, cells were arrested using Nocodazole (Sigma; SML1665, 50 ng/ml) for 2 h prior to metaphase spread preparation. EdU was detected using Click-iT Plus EdU labelling kits (Alexa Fluor 488, Invitrogen; C10637). Slides were air dried and counterstained using Vectashield with DAPI.

### Quantification and statistical analysis

Statistics analysis was performed by using GraphPad Prism software version 9 by two-tailed unpaired Student's $t$-test or Welch's $t$-test. Data

were presented as the mean ± standard deviation (S.D.) unless specified. Probability value "$p \leq 0.05$" is considered to be significant.

### Reporting summary

Further information on research design is available in the Nature Portfolio Reporting Summary linked to this article.

## Data availability

The datasets generated and/or analysed in the current study are available in the Source Data and in Figshare (https://doi.org/10.6084/m9.figshare.25818235). The mass spectrometry proteomics data have been deposited to the ProteomeXchange Consortium via the PRIDE69 partner repository with the dataset identifiers PXD064192, PXD052850, and PXD052669. Microscopy image data acquired in the current study are often associated with specific experimental setups, and the raw images and relevant details can be provided from the corresponding author upon request. Source data are provided with this paper.

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

## Acknowledgements

We would like to thank Professor Marcel van Vugt (University of Groningen) for the ΔBLM HAP1 cells, Professor Steve Jackson (University of Cambridge) for the ΔRIF1 RPE1 cell line, Professor Anne Donaldson and Dr. Shin-Ichiro Hiraga (University of Aberdeen) for the plasmids of EGFP-tagged human RIF1 WT and PP1bs, Professor Mark Burkard for the cDNA of CENPB (residue 1–158) and Dr Gary Ying-Wai Chan (the University of Hong Kong) for the CRISPR plasmids for tagging endogenous PICH with mAID-mClover3. We would like to acknowledge the BSRC Mass Spectrometry and Proteomics Facility (University of St. Andrews) and the Proteomics Core Facility (The Institute of Cancer Research) for the mass spectrometry services. We would like to thank Dr. Ramón González-Méndez from The Mass Spectrometry Research Facility (University of Sussex) for his help and technical support with in-house mass spectrometry. This research utilised Queen Mary's Apocrita HPC facility, supported by QMUL (Queen Mary University of London) Research-IT. We thank Jonathan Wing for the help with cell sorting (University of Sussex) and Dr. Yan Gu and the Wolfson Centre for Biological Imaging (University of Sussex) for providing excellent microscopy and imaging facilities. We thank Professors Aidan Doherty, Evi Soutoglou, Timothy Humphrey and the members of Chan Lab for their valuable comments on this manuscript. The current research was supported by the Sir Henry Dale Fellowship (104178/Z/14/A) provided by Wellcome Trust and the Royal Society and Wellcome Trust Career Development Award (225348/Z/22/Z) to K.L.C.

## Author contributions

M.F.C. and K.L.C. contributed to the conceptualisation and development of the project. M.F.C., U.A., E.K., T.O., A.W.O., A.C. and K.L.C. designed and executed the experiments and performed data analysis. A.D.H. developed the Fiji imaging analysis plugins. M.D. performed the AlphaFold2 structural predication and analysis. A.W.O. and M.D. generated recombinant CDK1/Cyclin B1 proteins. U.A. performed mass spectrometry analysis. M.F.C., E.K., U.A. and K.L.C. prepared figures and wrote the manuscript.

## Competing interests

The authors declare no competing interests.
