## [Transparent Peer Review file · Nature Communications]

Centromere protection requires strict mitotic inactivation of the Bloom syndrome helicase complex

Corresponding Author: Dr Kok-Lung Chan

Version 0:

Reviewer comments:

Reviewer #1

(Remarks to the Author)

The regulated removal of DNA-mediated linkages between chromosomes, particularly sister chromatids, during mitosis is an interesting and understudied area compared with the large number of studies on cohesin. In this work, Fernández-Casañas and co-workers explore the regulation of sister separation by the the BLM and TOP3A-containing BTRR complex during cell division. They report that the activity of this complex is suppressed by mitotic kinase activity in mitosis until the metaphase to anaphase transition. They propose that this regulation occurs at a number of levels, including recruitment, assembly, and activity of the complex. The authors use a number of elegant approaches to support their conclusions. Although the mechanistic details of how the different mitotic kinases each specifically regulate these different processes is not fully delineated here, the overall finding that mitotic kinases limit dissolvase activity in mitosis appears solid, and is useful in addressing some inconsistencies in the current literature. It is also interesting that controlling complex assembly appears to play an important role. This is different from models for other centromere activities which tend to emphasise the control of protein recruitment (eg for the CPC). The work also helps to understand the previously reported role of Plk1 in maintaining centromere chromatin integrity, and provides insight into a process that may underlie the formation of clinically-important chromosome aberrations such as whole arm translocations.

Major points

1. Although it is certainly plausible that Plk1 inhibition produces morphologically anaphase-like cells that are biochemically still in mitosis (ie Cdk1 is still active), cells such as those shown in Figure 1C, D are hard to distinguish from true anaphase cells. The use of Sgo1 RNAi will reduce anaphase entry to a large extent, but siRNA transfection is never 100%. It would be useful to have marker staining in experiments such as these to confirm that such cells are truly pre-anaphase. This is important because PICH and BLM are naturally recruited in anaphase.
2. In Figure S4C (BLM vs CENPB-RMI1) and in Figure 3D (TOP3A-7A vs CENPB-BLM) the two proteins show differences in their exact localizations. This is different from studies of the CPC, for example, where staining of the components of this complex (AurB, INCENP, Survivin, Borealin) are essentially indistinguishable. I wonder if this suggests that the observed recruitment is not as simple as BTRR complex formation?
3. BLM S144 is about as little like an optimal Mps1 site as it is possible to be (Mps1 is ranked 291 out of 303 Ser-Thr kinases as a likely kinase for this site based on the results of DOI: 10.1038/s41586-022-05575-3; see PhosphoSite Plus website). The original paper reporting this phosphorylation (Leng et al. 2006) does not provide any direct evidence for Mps1-mediated phosphorylation of S144 in vitro, and dependency in cells might well be due to indirect effects. Cdk1 has also been reported to phosphorylate S144. I think experiments that directly test the ability of Mps1 vs Cdk1 to phosphorylate BLM S144 in vitro and in cells are required before firm models can be proposed, at least with respect to the role of Mps1 activity.
4. Similarly, for the proposed Plk1 sites, for only one of these (S358) does Plk1 appear in the top 50 predicted kinases based on the results of DOI: 10.1038/s41586-022-05575-3; see PhosphoSite Plus website. I fully acknowledge that these are imprecise predictions, but it does raise the question of whether these are truly targets of the proposed kinase. Can additional evidence be provided?
5. Although it would not fully distinguish between direct and indirect effects of kinases on BLM, have the authors tested if E

or D mutants of key phosphosites can rescue the effects of Plk1, Mps1 or Cdk1 inhibition? Of course, negative results here would not rule out the authors' model (for example, D/E may not mimic phosphorylation sufficiently), but positive results would provide good support.

Minor Points

1. Figure 1D: please explain how "maximum chromatid separation distance" is measured.
2. Figure 2A. Will be too small to read in final figure. A similar problem likely applies to other figure parts.
3. Figure 6. I think the diagrams of the models are important for understanding the work, but I find these hard to follow. I think further thought should be given to these. The "BTRR assembly" box is adrift from the BLM – TOP3A/RMI1 interaction, and there is inconsistency in the use of arrows, for example. Mps1, Plk1 and Cdk1 are all kinases that are proposed to act on the BTRR complex, yet Plk1 has an arrow to its targets, Mps1 appears to bind to its proposed target (S144), and Cdk1 has no direct interface with BTRR in the figure. In the case of Cdk1, perhaps this is because these sites might be indirectly regulated by Cdk1 (see Discussion), but this all makes for a confusing figure.

Reviewer #2

(Remarks to the Author)

The group of Kok-Lung Chan identified an unexpected role of the kinase Plk-1 in protecting centromere integrity possibly by blocking the hyperactivity of the helicase BLM (PMID: 31253795). In this follow up work by Fernandez-Casañas et al. they investigated the regulation of the factors that promote maintenance of centromere stability during mitosis by studying more precisely the molecular mechanisms of this regulation. They followed up on previous works regarding the phosphorylation and rapid dephosphorylation of BLM and PICH in mitosis. They reveal that the BTRR proteins (BLM-TOP3A-RMI1-RMI2), a complex that normally acts in anaphase to resolve ultra-fine bridges, undergo to a define temporal inactivation via several kinases that destabilise its assembly and function. They discovered the phosphorylation regions on BLM and the role of the mitotic kinases and phosphatases such as Cdk1, Plk1 and PP1 to regulate BTRR activity, in what they define a mitotic centromere protection mechanism. They also show the kinetics of this temporal disassembly of the BTRR complex (when cells enter mitosis until they reach anaphase) and how this regulation is essential to avoid abnormal centromere decompaction and break.

In general, this is a very interesting manuscript, well written and clear. The figures are accurate and understandable. However, there are still some important points to be addressed before final acceptance of this work in Nat Com.

Major comments

- Figure 1A shows a Co-IP experiment in which cells are arrested in metaphase using MG132. This is a potent proteasome inhibitor. To rule out whether the increase in co-precipitated TOP3A/PICH may be dictated by the fact that the normal degradation process of protein is disrupted I would suggest repeating the same experiment in the presence of other inhibitors that induce mitotic arrest such as apcin. Also, the experiments should be done in triplicate.
- While the reviewer appreciates that quality of the work and the fact that many centromeres/kinetochores are analyzed is unclear how many times the experiments were done. This is true for most of the figures, except the last two figures where every experiments were done in triplicate. Having hundreds of centromeres/kinetochores analyzed in one experiment does not have the same biological meaning that analyzed less centromeres/kinetochores in ≥ 3 experiments. The authors should clarify this point and reproduce their data accordingly. This is a big flaw of the entire work.
- Figure 1C exploits SGO1 depletion to induce premature loss of sister chromatid cohesion. This is a confusing experiment that can lead to misinterpretation, especially due to the fact that Sgo1 was depleted for several days. It is unclear how the loss of cohesion is also induced since SGO1 is mainly specific to the centromere and not acting at the chromosome arms. There is also no demonstration that SGO1 is depleted via western blot or immunofluorescence.
- The movies of figure 3B showed 12'-20' time frame. I am not sure if these were the movies used to make the graph in 2B, but how do the authors manage with such low temporal resolution to distinguish all the different stages of mitosis that, with some exception, only last few minutes? This is particularly true for prophase-prometaphase-metaphase transition. Timing should also be reported on the pictures since Movies S1 and S2 have different time frames.
- The authors nicely show that mutations of seven residues in CENPB-mCherry-BLM (7A), rescued the loss of EGFP-TOP3A centromeric recruitment in early mitotic cells. The authors should better define which residues play a key role in this process by mutant them individually or in couples. From the prediction on Figure 4D it also seems that those residues are in contact with RMI1 and not only TOP3A, so the tail might be important for stabilization of the entire BTRR complex. The authors should check by IF or Co-IP if binding to RMI1 is also rescued.
- Line 343: "Further mutagenesis revealed a small cluster of PLK1 dependent phosphorylation sites (6A) locating in the middle of the flexible N-terminal region of BLM crucial for suppressing the illegitimate centromere unwinding". To this

reviewer it is not clear how they reach to this conclusion since in figure 4E they either mutate all 12 residues or only the 6 residues of the cluster. The authors should test the S>A mutants only in the 6 residues not considered part of the cluster to see if the phenotype is rescued and if they observe centromere unwinding.

- Line 358: "This was not due to the retention of PLK1-dependent hyper-phosphorylation (Fig. 5B)". This is an unclear statement and should be better explained. The authors should help the readers to interpret the results of the WB. Also, the blot needs proper quantification and replicas, not to mention that a loading control is missing.

Minor comments

- Can the authors define the term "high-precision microscopy"?
- Can the authors define what is a "CENPB peptide"? I guess they refer to the CENP-B DNA binding domain. If so, they need also to cite PMID: 1469042 that identified this domain.
- In figure 2D the author should change the Y-axis legend of the graph from centromere unwinding to RPA70 positive. Similarly, they need to clarify this is the main text.
- Although it is very clear from the text that for the experiment shown in Figure 3 a helicase-mutant form of BLM was used in order to avoid disruption of centromeric chromatin, only 'CENPB-mCherry-BLM' is indicated in Figure 3. This should be changed to CENPB-mCherry-BLM (Q672R).
- The quality of WB and FACS in figure S4A is poor. Please replace with high resolution images.
- Line 938 correct to air dried.
- There are several formatting errors that lead to some missing characters. Please check line 980, 993, 1002, 1013, 1023, 1034, Supp. Fig. 5 panel B close to Δ BLM on graph.
- Line 1067 correct to manufacturer's.
- Line 1078 correct to released.
- Figure 4E western blot is misplaced on the top of the IF picture. In addition the gel is cut on the top preventing clear analysis of the different EGFP-BLM mutants.
- Supplementary Figure 4D and 4E miss indication of pre-early-late-anaphase on pictures like it has been done in figure 1C.

Reviewer #3

(Remarks to the Author)

This manuscript describes the mechanism regulating the BTRR complex, comprising BLM, TOP3A and RIM1/2, focusing on its function at centromere-associated UFBs (marked by PICH) during mitosis. Using sophisticated microscopy approaches, Fernández-Casañas et al show that PLK1 inhibition induces the enrichment of BLM and RPA at centromeres. They then show the involvement of CDK1, PLK1/MPS1-mediated signalling and the antagonistic effect of RIF1-PP1 in this process. They propose that premature BLM occupancy at centromeres triggers illegitimate centromere unwinding prior to anaphase onset, which is prevented by the elaborate action of CDK, PLK1, and MPS1.

While this study is interesting and presents a very attractive model, I have several concerns which need to be addressed:

1. In this study, centromeric RPA is inferred as 'centromere unwinding' in cells mitotically arrested by MG132/PLK1i. However, it has previously been proposed that centromeric RPA is associated with R-loops and spindle assembly checkpoint (SAC) (PMID: 29170278). Could the authors rule out the possibility that BLM is alternatively involved in SAC? It would also be good to see some other independent evidence for 'centromere unwinding', e.g. ssDNA detection by BrdU labelling or FISH probe in non-denaturing conditions.
2. Based on the data shown in Figure 3A and B, the authors propose that the BLM-TOP3A interaction is down-regulated specifically at centromeres in early mitosis. It is worth noting that BLM-TOP3A also acts at other common fragile sites or telomeres-associated UFBs. To support the current model, the authors should test whether the BTRR complex actually remains at non-centromeric loci prior to anaphase onset. They could consider tethering BLM to telomeres by fusing it to a telomere-binding protein. Alternatively, a mammalian fluorescent two-hybrid (F2H) assay might be useful.
3. They mix and match non-cancerous diploid RPE1 cells (Figs 1 and 5), haploid HAP1 cells (Figs 2, 4, and phosphoproteomics) and cancerous HeLa cells (Fig 3) etc. They should justify the choice of cell line, reconcile their observations and discuss the biological implications more fully. Indeed, some of the inconsistencies between their findings and previous work could be explained by the difference in cell types, e.g. the effect of S144 on SCE between their system (using haploid HAP, therefore replication stress is lower than normal diploid cells) and the previous work (ref 18; PMID: 35119917) (using aneuploid HeLa cells, therefore increased replication stress) (lines 327-331).
4. It is unclear whether biological replicates were performed for experiments shown in main figures 1C, 2A, D, 3B, C, D, 5E, F and supplementary figures S1B, E, S2A, S3A, B, C, S4C and B. To reflect the reproducibility of their observation, statistical tests should also be performed between the median (or mean if normal distribution) of the biological replicates.

Formatting Figures:

1. 'STB' (e.g. Figure 1A): I think it stands for single thymidine block. Specify in the manuscript.
2. 'Centromere unwinding' (e.g. Figure 2D): it is more accurate to refer to this as 'RPA intensity'.
3. Panels with multiple results (e.g. Figure 2D, explained as top, middle and bottom etc): These should be split as it is not so

easy to follow when reading the manuscript.

4. Figure 3B. The number of cells evaluated seems very limited. Same issue for

5. Figure 3C. I don't quite understand what is meant by 'centromere clusters'. Have you analysed each centromere in a single cell and plotted them with several circles of the same colour? If so, the number of cells analysed is very limited.

5. Figure 4E needs to be corrected.

6. Figure S3 doesn't seem to include RPE1, although the figure legend says it does.

7. Figure S5A. One more biological repeat needs to be included.

Version 1:

Reviewer comments:

Reviewer #1

(Remarks to the Author)

Fernández-Casañas et al. have taken the reviewers' comments seriously, and have done an excellent job addressing the concerns. I am happy to commend the authors on an excellent piece of work, and to recommend publication of the paper in Nature Communications.

My only remaining suggestion would be to include discussion of the fact that the S144 site is a very poor match for the optimal Mps1 sequence (which bolsters their claims), and perhaps to comment on the extent to which the S144 and putative Plk1 sites match the optima for Cdk1 and Plk1 kinases. Also, on lines 539-540, they mention "another kinase(s), downstream of CDK1". Are these not the Plk1 sites they identify in the study?

Reviewer #2

(Remarks to the Author)

In the revised manuscript, Fernandez-Casañas et al. present additional experiments that significantly strengthen their previous conclusions regarding the molecular regulation of centromere stability during mitosis. They demonstrate that the BTRR complex (comprising BLM, TOP3A, RMI1, and RMI2), which functions in the resolution of ultra-fine DNA bridges during anaphase, undergoes temporally regulated disassembly via phosphorylation by mitotic kinases until anaphase onset. In particular, the roles of CDK1 and PLK1 as key mitotic kinases mediating BTRR disassembly have been further clarified. New experimental data identify Ser144, along with several PLK1 target sites on BLM, as critical phosphorylation sites.

Mutation of these residues results in premature activation of BLM's DNA unwinding activity at centromeres. The only caveat is that the earlier version of the manuscript proposed a regulatory axis involving MPS1 and PLK1 suggesting that MPS1 phosphorylates Ser144 on BLM (Figure 6 previous version of the manuscript). However, this new version completely revised this hypothesis. In vitro assays now implicate CDK1, rather than MPS1, as the kinase responsible for Ser144 phosphorylation. It is important to note that these phosphorylation studies were conducted using a short BLM peptide rather than the full-length protein. Despite it is sufficient to reveal phosphorylation by Cdk1, it might not be the case for Mps1, therefore not excluding a possible role of this kinase in Ser144 phosphorylation in vivo. In this regard, the authors have acknowledged this limitation in their conclusion.

All major and minor concerns raised by the reviewers have been comprehensively addressed. Revisions to figure labelling and legends have improved clarity and interpretability of the figures. Additionally, the number of replicates analysed is now well defined, enhancing confidence in the reproducibility of the results and the robustness of the proposed mechanism. The manuscript should now be accepted in Nature Communications.

Reviewer #3

(Remarks to the Author)

The manuscript is significantly improved with the addition of new experimental results and biological replicates. The work is now much more rigorous, and I found it a pleasure to read.

Based on the authors' response, I have one key request: to include ploidy data for the HAP1 cells used in this study, as this is important for confirming the reliability of the findings. Additionally, I offer a few suggestions to improve the clarity and precision of the manuscript.

Line 222 – The authors should provide more detail on the experimental model in the text, i.e. HAP1 WT and dBLM cells complemented with BLM variants, and present FACS profiles of HAP1 variants to confirm comparable ploidy levels in all. As the authors noted in their reply, WT HAP1 cells are expected to be haploid, but they are susceptible to becoming diploid. This must be properly monitored (PMID: 33184093).

Line 250 – The author should explicitly state that the DNA unwinding activity was 'inferred from centromeric RPA staining' (Ref. 20). While Ref. 20 and Fig. 2E support a correlation between BLM activity and RPA staining, the earlier publication from the same group does not directly substantiate this specific claim. In fact, Ref. 20 uses more cautious wording, stating that "the RPA association likely represents the presence of single-stranded DNA."

Lines 255-256 The sentence "we depleted TOP3A in HeLa and U2OS cells and found that also abolished BLM localisation to UFBs" is unclear. The referent of "that" is ambiguous.

Line 262. The authors should justify in the text why they switched to using HeLa cells, i.e. noting that these cells are more resistant to CENP-B overexpression (possibly due to a lack of p53 expression?).

Line 290. There is a redundancy in the sentence 'Repeatedly, we consistently...'

Lines 359-362 - HeLa cells are inherently genetically unstable, leading to substantial variability between cultures across different laboratories (PMID: 26483214, 30778230). Given the inconsistent findings reported in HeLa cells, the author's claim dismissing a role for mitotic BLM in suppressing crossovers appears overstated and should be moderated.

I suggest the authors limit their use of the word "believe", which appears six times (lines 228, 382, 408, 441, 560, and 129). Its frequent use lends a subjective tone that may detract from the manuscript's scientific objectivity.

Responses to reviewers' comments

We appreciate all the reviewers for their very positive comments and instructive suggestions on our manuscript. We have performed additional experiments as suggested and amended our manuscript accordingly. Overall, this has significantly strengthened our study and improved our final models. We also provided detailed explanations to address all reviewers' questions.

Reviewer #1 (Remarks to the Author):

The regulated removal of DNA-mediated linkages between chromosomes, particularly sister chromatids, during mitosis is an interesting and understudied area compared with the large number of studies on cohesin. In this work, Fernández-Casañas and co-workers explore the regulation of sister separation by the BLM and TOP3A-containing BTRR complex during cell division. They report that the activity of this complex is suppressed by mitotic kinase activity in mitosis until the metaphase to anaphase transition. They propose that this regulation occurs at a number of levels, including recruitment, assembly, and activity of the complex. The authors use a number of elegant approaches to support their conclusions. Although the mechanistic details of how the different mitotic kinases each specifically regulate these different processes is not fully delineated here, the overall finding that mitotic kinases limit dissolvase activity in mitosis appears solid, and is useful in addressing some inconsistencies in the current literature. It is also interesting that controlling complex assembly appears to play an important role. This is different from models for other centromere activities which tend to emphasise the control of protein recruitment (eg for the CPC). The work also helps to understand the previously reported role of Plk1 in maintaining centromere chromatin integrity, and provides insight into a process that may underlie the formation of clinically-important chromosome aberrations such as whole arm translocations.

We thank this reviewer for commenting that our overall findings are solid and useful in addressing inconsistencies in previously published studies. The new experiment of in vitro phosphorylation analysis suggested by this reviewer also helps us to challenge the previous study and improve our new model. We also thank the reviewer for pointing out that our study provides a different view and important mechanism on how human cells maintain centromere stability in contrast to research focusing on CPC biology.

Major points

1. Although it is certainly plausible that Plk1 inhibition produces morphologically anaphase-like cells that are biochemically still in mitosis (i.e. Cdk1 is still active), cells such as those shown in Figure 1C, D are hard to distinguish from true anaphase cells. The use of Sgo1 RNAi will reduce anaphase entry to a large extent, but siRNA transfection is never 100%. It would be useful to have marker staining in experiments such as these to confirm that such cells are truly pre-anaphase. This is important because PICH and BLM are naturally recruited in anaphase.

We thank for the suggestion to confirm that the anaphase-like cells observed in the PLK1i-treated SGO1-depleted cell population are not those that escaped into anaphase. As shown in Fig. 1E, we detected almost no normal anaphase cells after SGO1 knockdown in the DMSO conditions. This suggests that the SGO1 knockdown is very efficient in preventing mitotic exit or anaphase onset. Importantly, the emergence of the anaphase-like population is induced by PLK1i treatments, arguing against the insufficient of SGO1 knockdown. As

recommended by the reviewer, we have now also examined two “anaphase markers”: the loss of Cyclin B1 and midzone formation/localisation of INCENP and Aurora B to confirm that the anaphase-like cells remain stayed at pre-anaphase mitotic stages (New Figs. 1F, 1G and Supplementary Fig. 1F). Almost 100% of the anaphase-like cells maintain a high level of Cyclin B1, like the control (pro)metaphase cells (New Figs. 1F and Supplementary Fig. 1F; please also see the neighbouring G2/metaphase cells as internal staining controls for Cyclin B1 and Aurora B). In addition, we also show that they lack the midzone localisation of both INCENP and Aurora B CPC proteins (New 1G and Supplementary Fig. 1F). Together, we conclude that inhibiting PLK1 in the SGO1-knockdown mitotic cells does not trigger normal anaphase onset but facilitates separation of catenated chromatids mimicking anaphase cells. Therefore, we called this an “anaphase-like” pattern.

2. In Figure S4C (BLM vs CENPB-RMI1) and in Figure 3D (TOP3A-7A vs CENPB-BLM) the two proteins show differences in their exact localizations. This is different from studies of the CPC, for example, where staining of the components of this complex (Aruba, INCENP, Surviving, Borealin) are essentially indistinguishable. I wonder if this suggests that the observed recruitment is not as simple as BTRR complex formation?

Fig. S4C moved to Fig. S5; Fig. 3D moved to Fig. 3E.

We thank for this question as we also noticed the slight difference of the staining signals.

Under normal conditions, we never detect endogenous and ectopically expressed BTRR subunits localising to the K-chromatin and core centromeres. Only following acute PLK1/CDK1 inhibition does the BTRR complex start accumulating at K-chromatin. The core centromere regions also always lack the BTRR complex localisation in unperturbed mitotic cells.

We make use of this BTRR complex-free, centromeric zone to set up the *in vivo* tethering system to determine the interaction among the BTRR complex subunits during cell division. To efficiently drag the BTRR complex proteins to the core centromeres in early mitotic cells, it requires one of the subunits tagged with CENPB(1-158) and the inhibition of PLK1 or CDK1 (new Fig. S5). Alternatively, it can be also achieved by introducing phosphorylation mutations at the N-terminal part of BLM (new Fig. 3D). Generally, we observe very good co-localisation signals between the BTRR subunits, but we agree with the reviewer on this observation that they are never perfect. This could be due to the imbalance of protein expression (e.g. ectopic CENPB-mCherry-RMI1 overexpression vs. endogenous BLM). Besides, the signal of mCherry is always weaker than IF-stained proteins, which might influence the imaging.

As mentioned above, in the absence of the artificial CENPB-tag, none of the BTRR subunits can localise to the core region of the centromeres, no matter under what conditions. Based on this, we conclude that the reassembly of BTRR complexes at core centromeres mainly relies on the interaction between the subunits. Currently, we could not provide other possible explanations.

3. BLM S144 is about as little like an optimal Mps1 site as it is possible to be (Mps1 is ranked 291 out of 303 Ser-Thr kinases as a likely kinase for this site based on the results of DOI: 10.1038/s41586-022-05575-3; see PhosphoSite Plus website). The original paper reporting this phosphorylation (Leng et al. 2006) does not provide any direct evidence for Mps1-mediated phosphorylation of S144 in vitro, and dependency in cells might well be due to indirect effects. Cdk1 has also been reported to phosphorylate S144. I think experiments that directly test the ability of Mps1 vs Cdk1 to

phosphorylate BLM S144 *in vitro* and in cells are required before firm models can be proposed, at least with respect to the role of Mps1 activity.

We completely agree with this reviewer. Although Leng et al have showed that MPS1 can phosphorylate a BLM fragment (residues 1-159) *in vitro*, they never mapped the phosphorylation site(s). Besides, the loss of S144 phosphorylation after prolonged MPS1 inhibition in mitotic cells could be an indirect consequence of premature mitotic exit and the loss of CDK1 activity.

Currently, there is no method to test whether MPS1 can directly phosphorylate BLM at S144 *in vivo*. However, we have now purified a short BLM fragment (residue 121-166) and performed *in vitro* kinase assays using recombinant MPS1 and CDK1/CyclinB1 followed by mass spectrometry phosphorylation analysis. We could not detect S144 phosphorylation after prolonged MPS1 incubation, but we found that CDK1/CyclinB1 can do so. We have amended our manuscript and model according to this data.

4. Similarly, for the proposed Plk1 sites, for only one of these (S358) does Plk1 appear in the top 50 predicted kinases based on the results of DOI: 10.1038/s41586-022-05575-3; see PhosphoSite Plus website. I fully acknowledge that these are imprecise predictions, but it does raise the question of whether these are truly targets of the proposed kinase. Can additional evidence be provided?

We have also performed *in vitro* PLK1 kinase assays and mass spectrometry analysis on the purified BLM fragments that cover the putative PLK1 target sites. We detected 8 strong and 2 weak phosphorylation sites out of the 12 proposed PLK1 sites (new Fig. 5C & 5D; red labels). In addition, we also detect 12 extra sites that were shown to be mitotic cell specific previously (new Fig. 5C & 5D; black labels). Attempts were also made to include the S144 region to test if pre-treatments of CDK1 may enhance PLK1 phosphorylation. However, it was not possible to express longer N-terminal BLM fragments, which may be due to the highly unstructured nature. Nevertheless, we have demonstrated that PLK1 can directly target BLM *in vitro*, supporting previous *in vivo* phosphoproteomic analysis.

5. Although it would not fully distinguish between direct and indirect effects of kinases on BLM, have the authors tested if E or D mutants of key phosphosites can rescue the effects of Plk1, Mps1 or Cdk1 inhibition? Of course, negative results here would not rule out the authors' model (for example, D/E may not mimic phosphorylation sufficiently), but positive results would provide good support.

Our current findings indicate that phosphorylation of S144 acts to suppress mitotic BLM activity. We have now performed experiments using a S144E mutant, which, in principle, mimics S144 phosphorylation and inactivates BLM during mitosis. Interestingly, we detected a partial effect of the S144E mutation to suppress centromere unwinding (new Fig. 4D). As suggested by the reviewer, we also mentioned in the text that may be due to insufficient phosphorylation mimicking.

Minor Points

1. Figure 1D: please explain how "maximum chromatid separation distance" is measured.

After SGO1 knockdown, we observed two masses of sister chromatids separating towards to the opposite poles. This is more obvious upon acute PLK1 inhibition. The maximum chromatid separation distances are measured by the length of the outer separated

chromatids mass. We have put a “yellow double-arrow line” to indicate the maximum distance measured in the images shown in Fig. 1D. We also repeated the measurements three times.

2. Figure 2A. Will be too small to read in final figure. A similar problem likely applies to other figure parts.

We have now enlarged the sizes of most figures when possible.

3. Figure 6. I think the diagrams of the models are important for understanding the work, but I find these hard to follow. I think further thought should be given to these. The “BTRR assembly” box is adrift from the BLM – TOP3A/RMI1 interaction, and there is inconsistency in the use of arrows, for example. Mps1, Plk1 and Cdk1 are all kinases that are proposed to act on the BTRR complex, yet Plk1 has an arrow to its targets, Mps1 appears to bind to its proposed target (S144), and Cdk1 has no direct interface with BTRR in the figure. In the case of Cdk1, perhaps this is because these sites might be indirectly regulated by Cdk1 (see Discussion), but this all makes for a confusing figure.

We thank the reviewer for the comment. We have made new diagrams to illustrate the models accordingly. We simplified how the activities of CDK1, PLK1 and RIF1-PP1s differentially regulate the assembly and activity of the BTRR/UFB-binding complexes.

Reviewer #2 (Remarks to the Author):

The group of Kok-Lung Chan identified an unexpected role of the kinase Plk-1 in protecting centromere integrity possibly by blocking the hyperactivity of the helicase BLM (PMID: 31253795). In this follow up work by Fernandez-Casañas et al. they investigated the regulation of the factors that promote maintenance of centromere stability during mitosis by studying more precisely the molecular mechanisms of this regulation. They followed up on previous works regarding the phosphorylation and rapid dephosphorylation of BLM and PICH in mitosis. They reveal that the BTRR proteins (BLM-TOP3A-RMI1-RMI2), a complex that normally acts in anaphase to resolve ultra-fine bridges, undergo to a define temporal inactivation via several kinases that destabilise its assembly and function. They discovered the phosphorylation regions on BLM and the role of the mitotic kinases and phosphatases such as Cdk1, Plk1 and PP1 to regulate BTRR activity, in what they define a mitotic centromere protection mechanism. They also show the kinetics of this temporal disassembly of the BTRR complex (when cells enter mitosis until they reach anaphase) and how this regulation is essential to avoid abnormal centromere decompaction and break.

In general, this is a very interesting manuscript, well written and clear. The figures are accurate and understandable. However, there are still some important points to be addressed before final acceptance of this work in Nat Com.

We thank the reviewer for the positive comments on our manuscript. We have now addressed the points raised by the reviewer (please see below).

Major comments

- Figure 1A shows a Co-IP experiment in which cells are arrested in metaphase using MG132. This is a potent proteasome inhibitor. To rule out whether the increase in co-precipitated TOP3A/PICH may

be dictated by the fact that the normal degradation process of protein is disrupted I would suggest repeating the same experiment in the presence of other inhibitors that induce mitotic arrest such as apcin. Also, the experiments should be done in triplicate.

We apologise that the labels of this experiment were not clear. All cells released from the single thymidine block (STB) were treated with MG132 to induce metaphase arrest. We have corrected the labels (Fig. 1A and Fig. S1A). Given that all samples were pretreated with MG132, we conclude that the enhancement of the interaction is specifically caused by PLK1 inhibition (or the loss of PLK1-mediated phosphorylation). Under these conditions, we did not observe obvious changes in protein levels of the PLK1i-treated cells. Therefore, the increases in the Co-IP efficiency are unlikely attributed to protein level differences.

We have now included three additional Co-IP experiments using nocodazole-arrested prometaphase cells and obtained the same results (Fig. 1A and Fig. S1A). Together with our cellular localisation analyses, we believe our data is strong enough to conclude that the PLK1-mediated phosphorylation destabilises the mitotic UFB-binding complex formation.

We would like to emphasise that, unlikely many previous studies using ectopically overexpressed PICH proteins, we performed Co-IP assays on endogenous proteins, which are expressed in a physiological level. Although this made our Co-IP experiments more difficult to perform, it avoids potential artifacts as PICH overexpression usually leads to severe protein aggregation (PMID: 17218258 and unpublished results). This could interfere proper determination of protein-protein interaction between PICH and its binding partners. Therefore, we believe our approach reflects a more reliable result of the effect of PLK1 on the changes of the UFB-binding complex formation.

- While the reviewer appreciates that quality of the work and the fact that many centromeres/kinetochores are analyzed is unclear how many times the experiments were done. This is true for most of the figures, except the last two figures where every experiments were done in triplicate. Having hundreds of centromeres/kinetochores analyzed in one experiment does not have the same biological meaning that analyzed less centromeres/kinetochores in ≥ 3 experiments. The authors should clarify this point and reproduce their data accordingly. This is a big flaw of the entire work.

We appreciated the point raised by the reviewer regarding experimental repetitions. In addition to other new experiments as requested, we have now shown at least three independent repeats for the following experiments. We focused our efforts mainly on these analyses because we believe they are more critical and essential.

They include **Figs. 1A, 1C, 1D, 2B (previous 2A), 2E (previous 2D), 3C, 4D, 6D (previous 5E), S1A, S4A, S4C (previous 4B), S5 (previous S4C), S6A (previous S5A) and S7A (previous 5B).**

Previous Fig. S1E has been deleted as new Fig. 1C now shows three repeats of BLM staining on pre-anaphase and anaphase UFBs. The results are reproducible and consistent.

In Fig. S2A, we provided two repeated experimental counting (except in Q672R and 1-1333 mutants). We observed that various truncations or mutations at the very N-terminus of BLM can impair UFB localisation. Since multiple mutants defective in the TOP3A binding ($\Delta 51$, $\Delta 13$, N7A N8A, L9A, Q12A L13A) showed the same findings of the loss of UFB-localisation. Together with the Alpha-Fold2 model and more repeated analyses shown in Figs. 2B, 2E

and 2F, we did not pursue further repetition of this transient transfection experiment because the conclusion is very clear.

In Fig. S3, we carefully measured the dependence between BLM and TOP3A for UFBs localisation in four different cell lines. Although the data were from one experiment in each cell line, we observed the same results, which show that either ablation of BLM or TOP3A impair the other to localise to UFBs. More importantly, we also demonstrated that disrupting the TOP3A-binding interfaces impairs BLM localisation to both UFBs and K-chromatin as well as compromises the BTRR complex mitotic activity in repeated experiments (Figs. 2B, 2C, 2E, 2F and Fig. S2A). We believe our overall analyses are adequate and strong enough to conclude that the inter-dependence of BTRR complex subunits for UFB localisation and activity. Furthermore, our data is also in agreement with a previous study showing the necessity of RMI2 for BLM and TOP3A to localise to UFBs (PMID: 27977684; Figs. 6 & 7). Therefore, we did not pursue further repeated analysis.

We would like to emphasise that in the previous study (PMID: 30177760; Fig. 3), which they claimed that BLM and TOP3A can independently bind to PICH on anaphase UFBs, was based on single microscopy image without any quantitative measurements. There were no anaphase cell counting or UFB intensity measurements. There was also no indication of any contrast processing. Therefore, we believe our overall investigation is much more rigorous.

In Fig. 6E (previous Fig. 5F), we have used two different doses of the PP1s inhibitor to treat two different cell lines independently, and we obtained similar results. We thus believe the effect is reproducible and not due to technical errors.

- Figure 1C exploits SGO1 depletion to induce premature loss of sister chromatid cohesion. This is a confusing experiment that can lead to misinterpretation, especially due to the fact that Sgo1 was depleted for several days. It is unclear how the loss of cohesion is also induced since SGO1 is mainly specific to the centromere and not acting at the chromosome arms. There is also no demonstration that SGO1 is depleted via western blot or immunofluorescence.

We apologised the unclear labels in the experimental setup shown in Fig. 1C. The procedure was shown on the top of Fig. 1C. The Sgo1-depletion were carried out while cells were under single thymidine block (STB) and release. In total, the SGO1 RNAi only lasted for 33 hours but not several days. We aimed to deplete Sgo1 before the first mitosis following the STB release. We believe that the SGO1 RNAi efficiency was very high as we rarely observe anaphase(-like) cells (Fig. 1E; +siSGO1+DMSO). Since the SGO1 RNAi phenotype and effect were so obvious, we did not think a Western blot is needed. We have now provided a Western blot shown in new Fig. S1D.

SGO1 is crucial to protect sister-chromatids cohesion at centromere and pericentric regions. It functions to counteract the cohesin unloader, Wapl (PMID: 22901742; PMID: 23361318; PMID: 21111234), which mediates a prophase pathway to release cohesin from the chromosomal arms. Therefore, when normal cells enter mitosis and especially under prolonged mitotic arrest, sister chromatids cohesion largely loses at the chromatid arms first in a Wapl-dependent manner but not at pericentric regions because of the protection by SGO1. As a result, chromosomes manifest as X-shape chromosomes. After SGO1-depletion, Wapl can now also unload the cohesin from the pericentric regions, leading to complete premature sister chromatids separation. Since our cells were pre-treated with low doses of ICRF193 to induce catenations, the separation of sister chromatids thus looked more disorganised (Fig. 1E: +siSGO1+DMSO). The most interesting result is that the

addition of PLK1i can promote anaphase-like chromosome separation, presumably due to the activation of the BTRR/UFB-binding complexes to unwind the catenated DNA bridges.

- The movies of figure 3B showed 12'-20' time frame. I am not sure if these were the movies used to make the graph in 2B (we believe the reviewer referred to Fig. 3B), but how do the authors manage with such low temporal resolution to distinguish all the different stages of mitosis that, with some exception, only last few minutes? This is particularly true for prophase-prometaphase-metaphase transition. Timing should also be reported on the pictures since Movies S1 and S2 have different time frames.

The movie montages shown in Fig. 3B were generated from movies S1 and S2. Selected timepoints were chosen to show representative examples of different stages of mitosis and the G2 phase beforehand as well as the subsequent G1. We have now put the time stamps in the live-cell images.

The reviewer is correct that mitosis is normally last for ~60-70min (e.g. ~10min prophase; ~15-20min prometaphase; ~10-15min metaphase; ~15min anaphase/telophase). However, we found that the expression of the CENPB-mCherry-RMI1 and -BLM(Q672R) fusion proteins can prolong mitosis, especially during prometaphase and metaphase. Therefore, we re-adjust our imaging intervals (12min or 20min intervals) in order to capture the entire mitosis/cell division process. This also helped reduce potential phototoxicity. Moreover, individual mitotic cells have different prometaphase and metaphase durations, so we could not align the intensity profiles merely based on time. Instead, we grouped cells according to their mitotic morphologies and stages, and classified them as different mitotic stages. A timeframe before the earliest prometaphase image is considered as prophase because we could not detect any elongation of prophase. Likewise, the timeframe before prophase is considered as late G2. It is worth to mention that the time-lapse imaging experiments were done by monitoring lots of random individual interphase cells. They may or may not be at G2 phase when the live-cell imaging begun. Many of the interphase cells did not enter mitosis during the entire imaging periods. Therefore, this limits the total numbers of cells for analysis. However, we must emphasise that our system shows that nearly all individual cells tracked exhibit the same patterns of the dynamic changes in the TOP3A-BLM interaction; namely a transient loss of protein-protein interaction during early mitosis and a restoration upon mitotic exit. This transient protein-protein interaction loss was not detected in between TOP3A and RMI1 in all individual cells examined.

- The authors nicely show that mutations of seven residues in CENPB-mCherry-BLM (7A), rescued the loss of EGFP-TOP3A centromeric recruitment in early mitotic cells. The authors should better define which residues play a key role in this process by mutant them individually or in couples. From the prediction on Figure 4D (we believe the reviewer referred to Fig. 3D) it also seems that those residues are in contact with RMI1 and not only TOP3A, so the tail might be important for stabilization of the entire BTRR complex. The authors should check by IF or Co-IP if binding to RMI1 is also rescued.

We thank the reviewer for the suggested experiments. The aim of this experiment was to test if potential phosphorylation at the residues surrounding the TOP3A/RMI1 binding interfaces of BLM may destabilise their interaction. We have now included two additional mutants (6A) and S17A, T20A, which both can restore the TOP3A recruitment to by the CENPB-mCherry-BLM tethering system.

We and other have shown that TOP3A and RMI1 constitutively interact with each other throughout the cell cycle. Importantly, the interaction is not disrupted during mitosis (Fig. 3B). Therefore, it is hard to imagine that the recruitment rescue by the BLM phosphomutants only occurs for TOP3A but not RMI1. Since these two proteins always maintain stable interaction or exist as a stable heterocomplex, we find no strong reason to check the RMI1 recruitment, which is very likely following the same pattern as TOP3A does.

- Line 343: "Further mutagenesis revealed a small cluster of PLK1 dependent phosphorylation sites (6A) locating in the middle of the flexible N-terminal region of BLM crucial for suppressing the illegitimate centromere unwinding". To this reviewer it is not clear how they reach to this conclusion since in figure 4E they either mutate all 12 residues or only the 6 residues of the cluster. The authors should test the S>A mutants only in the 6 residues not considered part of the cluster to see if the phenotype is rescued and if they observe centromere unwinding.

We thank the reviewer for this important experiment.

We have now shown that mutations at the other 6 residues outside the cluster also have a dominantly active effect on BLM to unwind centromeric DNA (new Fig. 4E). We thus have changed our interpretation and proposed that phosphorylation at multiple sites is probably required to completely suppress BLM activity. This may also help explain why BLM is always hyper-phosphorylated in early mitosis. In addition, we also provide new data of *in vitro* PLK1 kinase assays (new Fig. 5) showing 10 out of 12 the proposed sites can be directly phosphorylated by recombinant PLK1.

- Line 358: "This was not due to the retention of PLK1-dependent hyper-phosphorylation (Fig. 5B)". This is an unclear statement and should be better explained. The authors should help the readers to interpret the results of the WB. Also, the blot needs proper quantification and replicas, not to mention that a loading control is missing.

We show that in the absence of RIF1, the loading of BTRR complexes at K-chromatin and the induction of centromere unwinding induced by PLK1 inhibition become less efficient. We thus questioned whether this may be due to inefficient removal of the PLK1-mediated hyperphosphorylation on PICH and BLM proteins. Thus, the Western blot was used to determine whether the hyper-phosphorylated forms of BLM and PICH remain after PLK1i treatments in the absence of RIF1 (Fig. 5B moved to new Fig. S7A with three repeated experiments). The answer is clearly not. We observed 100% loss of the hyper-phosphorylated forms of BLM and PICH irrespective of RIF1.

We did not show a loading control because in this type of analysis, we were comparing the hyper-phosphorylated, partial-phosphorylated and non-phosphorylated forms of the proteins. The loading control is not crucial for the interpretation of the phospho-shift results. Nevertheless, we included Ku80 as a loading control for the latest two repeated experiments.

Minor comments

- Can the authors define the term "high-precision microscopy"?

We regularly use 200nm diameter of TetraSpeck beads to calibrate and align our microscope systems to ensure that our imaging data were not interfered by chromatic aberration (please see our previous study PMID: 31253795, Supplementary Fig. 11).

Therefore, this allows us to confidently determine the precise (relative) positions of protein foci labelled by different fluorophores. More importantly, we observed a “mirror” pattern of PICH localisation underneath the inner kinetochores of the sister centromeres, which clearly reveals the K-chromatin compartment. We thus called our method as high-precision microscopy.

We have also added a description to the Method section.

- Can the authors define what is a “CENPB peptide”? I guess they refer to the CENP-B DNA binding domain. If so, they need also to cite PMID: 1469042 that identified this domain.

Yes, this is correct. We used the first 158 residues of CENPB that binds to the 17bp repeats of CENP-B box. We have now changed “CENPB peptide” to “truncated CENPB (residues 1-158)” and cited the reference (PMID: 1469042).

- In figure 2D the author should change the Y-axis legend of the graph from centromere unwinding to RPA70 positive. Similarly, they need to clarify this is the main text.

Fig. 2D moved to Fig. 2E. We have changed to “% mitotic cells positive for centromeric RPA binding”. We have mentioned in the text that the centromeric RPA foci/threads formation requires active BLM DNA unwinding activity (See Fig. 2E and our previous study PMID: 31253795, Figs. 6C & 6D). A helicase-dead BLM (Q672) can no longer able to trigger centromeric RPA foci/threads despite its proficiency in K-chromatin localisation after PLK1i treatments. Therefore, the presence of centromeric RPA is not a result of simple BLM-RPA interaction but dependent of active DNA unwinding. We thus use it as a surrogate marker for centromeric DNA unwinding throughout the study.

- Although it is very clear from the text that for the experiment shown in Figure 3 a helicase-mutant form of BLM was used in order to avoid disruption of centromeric chromatin, only ‘CENPB-mCherry-BLM’ is indicated in Figure 3. This should be changed to CENPB-mCherry-BLM (Q672R).

Changed.

- The quality of WB and FACS in figure S4A is poor. Please replace with high resolution images.

Figs S4A and S4B have been moved to Figs. S4B and S4C.

Previous Figs. S4A and S4B were prepared in the same way, and we were not sure what might cause the quality change of the image in Fig. S4A but we have re-generated the figure.

- Line 938 correct to air dried.

Corrected.

- There are several formatting errors that lead to some missing characters. Please check line 980, 993, 1002, 1013, 1023, 1034, Supp. Fig. 5 panel B close to Δ BLM on graph?.

Corrected.

- Line 1067 correct to manufacturer’s.

Corrected.

- Line 1078 correct to released.

Corrected.

- Figure 4E western blot is misplaced on the top of the IF picture.

Corrected. We have deleted the representative IF images to save the space.

In addition the gel is cut on the top preventing clear analysis of the different EGFP-BLM mutants.

A new gel image containing additional mutant 6A-2 was shown.

- Supplementary Figure 4D and 4E miss indication of pre-early-late-anaphase on pictures like it has been done in figure 1C.

We believe the reviewer mentioning Fig. S1D and E.

Fig. S1D has been moved to Fig. S1E and the labels were added. We have deleted Fig. 1E as it showed similar results as in the new Fig. 1C.

Reviewer #3 (Remarks to the Author):

This manuscript describes the mechanism regulating the BTRR complex, comprising BLM, TOP3A and RIM1/2, focusing on its function at centromere-associated UFBs (marked by PICH) during mitosis. Using sophisticated microscopy approaches, Fernández-Casañas et al show that PLK1 inhibition induces the enrichment of BLM and RPA at centromeres. They then show the involvement of CDK1, PLK1/MPS1-mediated signalling and the antagonistic effect of RIF1-PP1 in this process. They propose that premature BLM occupancy at centromeres triggers illegitimate centromere unwinding prior to anaphase onset, which is prevented by the elaborate action of CDK, PLK1, and MPS1.

While this study is interesting and presents a very attractive model, I have several concerns which need to be addressed:

We thank this reviewer for finding our study interesting. We have now addressed the points raised by the reviewer.

1. In this study, centromeric RPA is inferred as 'centromere unwinding' in cells mitotically arrested by MG132/PLK1i. However, it has previously been proposed that centromeric RPA is associated with R-loops and spindle assembly checkpoint (SAC) (PMID: 29170278). Could the authors rule out the possibility that BLM is alternatively involved in SAC? It would also be good to see some other independent evidence for 'centromere unwinding', e.g. ssDNA detection by BrdU labelling or FISH probe in non-denaturing conditions.

We thank the reviewer for the question and suggestion. A previous study (PMID: 29170278) has claimed that ATR is recruited to mitotic centromeres/kinetochores through an Aurora A-

CENPF dependant pathway, which then phosphorylates the RPA that binds to centromeric R-loops for SAC controls.

Under normal mitotic conditions, we never detect any RPA signals at the centromeres/K-chromatin in all examined cell lines. The RPA foci/threads observed throughout our study only detectable after the inhibition PLK1 or CDK1, whereas the latter shows lower efficiency. Most importantly, we show that the K-chromatin/centromeric RPA foci are completely dependent on BLM's helicase activity (See Fig. 2E and our previous study PMID: 31253795, Figs. 6C & 6D). Therefore, the centromeric RPA signals detected in our study serves as a surrogate marker of unwound DNA. Given that this is strongly dependent of BLM activity and can give rise to centromere disintegration, we don't think it is necessary to employ other methods for ssDNA detection.

Regarding the role of BLM on SAC, although a previous study (PMID: 16864798) has proposed that BLM S144 phosphorylation is required for SAC activation, we never detect any defects of mitotic arrest in BLM knockout, S144A and BS cell lines induced by nocodazole (data not shown). We find no evidence to believe that BLM plays a function in SAC. Whether BLM may influence R-loop formation at centromeres, we currently don't know but these other proposed roles won't influence the conclusion of our study, namely that mitotic hyperphosphorylation can restrict BLM to unwind centromeric DNA for the protection of centromere integrity.

2. Based on the data shown in Figure 3A and B, the authors propose that the BLM-TOP3A interaction is down-regulated specifically at centromeres in early mitosis. It is worth noting that BLM-TOP3A also acts at other common fragile sites or telomeres-associated UBFs. To support the current model, the authors should test whether the BTRR complex actually remains at non-centromeric loci prior to anaphase onset. They could consider tethering BLM to telomeres by fusing it to a telomere-binding protein. Alternatively, a mammalian fluorescent two-hybrid (F2H) assay might be useful.

We would like to clarify that the assays set up in Figs. 3A and B were to examine protein-protein interaction *in vivo* between different subunits of the BTRR complex and whether this is in a cell cycle dependent manner.

The reason we look at centromeres is because the loss of PLK1 activity leads to illegitimate assembly of the BTRR complex specifically at K-chromatin/centromeres, where PICH pre-occupied. This assembly does not occur at common fragile sites or telomeres because PICH does not bind to those loci unless there are unresolved UFBs exposed during anaphase.

Nevertheless, we have generated HeLa cells stably expressing a telomere-tethering fusion protein of RMI1 (New Fig. S4A) to test if the destabilisation of BLM and RMI1 interaction can also occur outside the centromere during mitosis. The answer is yes. This indicates the destabilisation of protein-protein interaction is mitosis, rather than centromere, specific. It is also consistent with the co-IP data shown in Figs. S4B and S4C, where weakened BLM-TOP3A interaction was detected from mitotic cells extracts and under high salt concentrations.

3. They mix and match non-cancerous diploid RPE1 cells (Figs 1 and 5), haploid HAP1 cells (Figs 2, 4, and phosphoproteomics) and cancerous HeLa cells (Fig 3) etc. They should justify the choice of cell line, reconcile their observations and discuss the biological implications more fully.

We have previously shown that acute PLK1 inhibition in human mitotic cells causes centromere disintegration in a BLM helicase activity-dependent manner. We find that different cell lines exhibit different frequencies of centromere disintegration but RPE1, HCT116 and HAP1 cells are particularly susceptible. We thus employed different cell lines

because they have different advantages and disadvantages to address particular biological questions.

RPE1, HAP1 cells and their derivatives are most suitable to study the conditions of how different kinases and phosphatases trigger the mitotic BTRR/UFB-binding complex activities. HAP1 Δ BLM cells allow us to study the effect of different BLM variants. They can stay longer at the metaphase-arrested stage after the administration of CDK1i than RPE1 cells. This gives a better window to study the initiation of K-chromatin unwinding.

On the other hand, HeLa cells (containing the CENPB fusion protein) are particularly useful for studying the dynamic protein-protein interaction *in vivo* throughout cell division. This is because their mitotic progression is less sensitive than in RPE1 cells after the expression of CENPB-fusion proteins.

U2OS, HeLa and its derivatives were used for Co-IP experiments because of their relatively high abundance of BLM and TOP3A proteins. Therefore, we carefully design and utilise the suitable cell lines for each experiment.

Indeed, some of the inconsistencies between their findings and previous work could be explained by the difference in cell types, e.g. the effect of S144 on SCE between their system (using haploid HAP, therefore replication stress is lower than normal diploid cells) and the previous work (ref 18; PMID: 35119917) (using aneuploid HeLa cells, therefore increased replication stress) (lines 327-331).

HAP1 cells get diploidised very quickly. All our HAP1 and HAP1 derived cells are diploid. Whether the previous observation of elevated SCEs in the BLM S144A HeLa cells (PMID: 35119917) are because of increased replication stress are difficult to determine. Notably, it was not only our current study showing the lack of increased SCEs in the BLM S144A cells, another study (PMID: 16864798) using Bloom's cells complemented with S144A mutant also failed to detect elevated SCEs. In addition, Andrew Blackford also informed us that RPE1 BLM Δ cells expressing S144A show no defects in SCE suppression (personal communication). Therefore, there is a lack of strong evidence to support that S144A is a loss-of-function mutation. In contrast, our current study clearly demonstrates that the S144A becomes dominantly active in driving K-chromatin unwinding during mitosis. We have included a short discussion in the Result section stating that the high SCEs in S144A HeLa cells shown previously may be a cell line specific phenomenon.

4. It is unclear whether biological replicates were performed for experiments shown in main figures 1C, 2A, D, 3B, C, D, 5E, F and supplementary figures S1B, E, S2A, S3A, B, C, S4C and B. To reflect the reproducibility of their observation, statistical tests should also be performed between the median (or mean if normal distribution) of the biological replicates.

We understood the reviewer's concern on experiment reproducibility. We agreed that that some of the experiments need repetitions and also provided reasons why some are not.

Fig. 1C, three repeats done.

Fig. 2A, moved to Fig. 2B, three repeats done.

Fig. 2D, moved to Figs. 2E and 2F, three repeats done.

Fig. 3B, this is a time-lapse single-cell measurement experiment. The beauty of this type of analysis is that all internal controls are present within the same examined condition. The aim of this experiment was to compare the protein-protein interaction dynamics of a cell going

through different cell cycle stages in real time within the same sample. We did not compare between samples. For instances, the CENPB-mCherry signals serve as an internal reference channel, showing the presence of a constant tethering of the “bait” at centromeres. The G2 and G1 phases of the same cell serve for internal comparison to the mitotic stages. The curve of each cell analysed show almost the same dynamic trend. Since we are not to compare between treatments, it is hard to explain that the transient loss of TOP3A-BLM interaction at the centromeres might be due to technical errors, and why so specific to only early mitosis. We have now topped up the number of cells analysed in the CENPB-mCherry-RMI1 cells, in which we could not detect any transient loss of the TOP3A-RMI1 interaction. The live-cell data of centromeric interaction between TOP3A and BLM(Q672R) was obtained from two independent experiments using two different clones. Importantly, the result of the live-cell analysis is further supported by the data shown in Fig. 3C. Therefore, we are very confident that there is a transient disruption of the BLM and TRR subcomplex interaction.

Fig. 3C, three repeats have been done. However, similar to the explanation for Fig. 3B, we did not think the repetition is critical in this type of analysis because we are comparing IF signals among different cell populations “within the same sample”. The experiment was not designed to compare between samples and/or treatments. Like the experiment in Fig. 3B, all internal references are present. It is hard to explain why the BLM centromeric recruitment drops only in the early mitotic cell populations, e.g. (pro)metaphase, versus the neighbouring interphase cells. This result is consistent with the live-cell imaging analysis in Fig. 3B.

Fig. 3D, three repeats done together with two extra mutants.

Fig. 5E, moved to Fig. 6E, three repeats done.

Fig. 5F, these experiments were done in two different cell lines using two different doses of the PP1s inhibitor. We thus did not purchase further analysis.

Fig. S1B, more measurements have been carried out in our previously study (PMID: 31253795 Fig. 4d)

Fig. S1E, we have removed this because we have repeated the same experiment in Fig. 1C.

Fig. S2A, we repeated once because multiple N-terminal mutants of BLM are found to have the same defects in UFB localisation. More analyses have also been carried out in Figs. 2B and 2C. Together with the AF2 model, all these results clearly show the defects of TOP3A binding compromising UFB localisation.

Fig. S3, we carefully measured the dependence between BLM and TOP3A for UFBs localisation in four different cell lines. Although the data were from one experiment in each cell line, we observed the same results, which show that either ablation of BLM or TOP3A impair the other to localise to UFBs. It is difficult to explain as a technical error. More importantly, we also demonstrated that disrupting the TOP3A-binding interfaces impairs BLM localisation to UFBs and K-chromatin as well as compromises the BTRR complex mitotic activity in repeated experiments (Figs. 2B, 2C, 2E, 2F and Fig. S2A). We believe our overall analyses are adequate and strong enough to conclude that the inter-dependence of BTRR complex subunits for UFB localisation and activity. Moreover, our data is also in agreement with a previous study showing defects of UFBs binding of BLM and TOP3A in cells lacking RMI2 (PMID: 27977684; Figs. 6 & 7). Thus, we did not think it is necessary to pursue further repeated analysis. I would like to also emphasise that in this previous study (PMID: 30177760; Fig. 3), where they claimed that BLM and TOP3A can independently bind to PICH-coated UFBs in anaphase cells, was based on single microscopy images without any quantitative measurements. There were no cell counting or UFB intensity measurements at

all. There was also no indication of contrast processing. Therefore, we believe our overall analyses are more rigorous and reliable.

Fig. S4C, moved to Fig. S5, three repeats done.

Fig. S4B, moved to Fig. S4C, three repeats done.

Formatting Figures:

1. 'STB' (e.g. Figure 1A): I think it stands for single thymidine block. Specify in the manuscript.

Done

2. 'Centromere unwinding' (e.g. Figure 2D): it is more accurate to refer to this as 'RPA intensity'.

Changed to “% mitotic cells positive for centromeric RPA binding”. We would like to emphasise that the centromere RPA signals detected are totally dependent of BLM helicase activity but not its physical occupancy. Therefore, we believe the centromeric RPA represents unwound ssDNA.

3. Panels with multiple results (e.g. Figure 2D, explained as top, middle and bottom etc): These should be split as it is not so easy to follow when reading the manuscript.

Split into Figs. 2E and 2F.

4. Figure 3B. The number of cells evaluated seems very limited. Same issue for

We have topped up the total numbers of cells analysed (n=16). This is single cell analysis and, as mentioned above, we observed almost all cells analysed follow the same pattern of the protein-protein interaction dynamics from G2 to mitosis and G1. Since all internal controls and references are provided within the same setup, we cannot find a reason to believe that these findings are due to technical issues. Furthermore, repeated analysis shown in Fig. 3C shows a consistent result.

5. Figure 3C. I don't quite understand what is meant by 'centromere clusters'. Have you analysed each centromere in a single cell and plotted them with several circles of the same colour? If so, the number of cells analysed is very limited.

We measured signal intensities at all single and overlapping centromeres and called them as centromere clusters. Now we present the data as “average centromere intensity per cell”. Three repeated measurements have been done from at least 50 cells.

5. Figure 4E needs to be corrected.

Corrected

6. Figure S3 doesn't seem to include RPE1, although the figure legend says it does.

We have also used RPE1 cells for the analysis but found out that depletion of TOP3A in RPE1 cells can lead to BLM degradation (data not shown). This has precluded our UFB localisation analysis. We have now removed RPE1 in the legend.

7. Figure S5A. One more biological repeat needs to be included.

Moved to Fig. S6A, three repeats done.

Responses to the reviewers' comments

Reviewer #1 (Remarks to the Author):

Fernández-Casañas et al. have taken the reviewers' comments seriously, and have done an excellent job addressing the concerns. I am happy to commend the authors on an excellent piece of work, and to recommend publication of the paper in Nature Communications.

My only remaining suggestion would be to include discussion of the fact that the S144 site is a very poor match for the optimal Mps1 sequence (which bolsters their claims), and perhaps to comment on the extent to which the S144 and putative Plk1 sites match the optima for Cdk1 and Plk1 kinases.

We thank this reviewer for recommending our study for publication and the suggestion.

We have now included a discussion and reference stating that the S144 sequence has very low substrate specificity for MPS1 according to a previous study. "Our finding also aligns well with the previous research showing very low substrate specificity of the S144 sequence for MPS1⁵²."

On the other hand, since we have experimentally proven that S144 containing region and the putative PLK1 sites can be efficiently targeted by CDK1 and PLK1, respectively, in vitro, we think it is not critical to comment their sequence specificities for these kinases.

Also, on lines 539-540, they mention "another kinase(s), downstream of CDK1". Are these not the Plk1 sites they identify in the study?

Near the TOP3A-binding interface, only two mitotic phosphorylation sites S17 and T20 have been reported. Since our mass spectrometry analysis did not pick up these two sites, it remains unclear whether they can be targeted by PLK1. Therefore, we used "another kinase(s),..." instead of PLK1.

Reviewer #2 (Remarks to the Author):

In the revised manuscript, Fernandez-Casañas et al. present additional experiments that significantly strengthen their previous conclusions regarding the molecular regulation of centromere stability during mitosis. They demonstrate that the BTRR complex (comprising BLM, TOP3A, RMI1, and RMI2), which functions in the resolution of ultra-fine DNA bridges during anaphase, undergoes temporally regulated disassembly via phosphorylation by mitotic kinases until anaphase onset.

In particular, the roles of CDK1 and PLK1 as key mitotic kinases mediating BTRR disassembly have been further clarified. New experimental data identify Ser144, along with several PLK1 target sites on BLM, as critical phosphorylation sites. Mutation of these residues results in premature activation of BLM's DNA unwinding activity at centromeres.

The only caveat is that the earlier version of the manuscript proposed a regulatory axis involving MPS1 and PLK1 suggesting that MPS1 phosphorylates Ser144 on BLM (Figure 6 previous version of the manuscript). However, this new version completely revised this hypothesis. In vitro assays now implicate CDK1, rather than MPS1, as the kinase responsible for Ser144 phosphorylation. It is important to note that these phosphorylation studies were conducted using a short BLM peptide rather than the full-length protein. Despite it is sufficient to reveal phosphorylation by Cdk1, it might not be the case for Mps1, therefore not excluding a possible role of this kinase in Ser144

phosphorylation in vivo. In this regard, the authors have acknowledged this limitation in their conclusion.

All major and minor concerns raised by the reviewers have been comprehensively addressed. Revisions to figure labelling and legends have improved clarity and interpretability of the figures. Additionally, the number of replicates analysed is now well defined, enhancing confidence in the reproducibility of the results and the robustness of the proposed mechanism. The manuscript should now be accepted in Nature Communications.

We thank this reviewer for recommending our study for publication.

Reviewer #3 (Remarks to the Author):

The manuscript is significantly improved with the addition of new experimental results and biological replicates. The work is now much more rigorous, and I found it a pleasure to read.

We thank you this reviewer for the very positive comment.

Based on the authors' response, I have one key request: to include ploidy data for the HAP1 cells used in this study, as this is important for confirming the reliability of the findings. Additionally, I offer a few suggestions to improve the clarity and precision of the manuscript.

Line 222 – The authors should provide more detail on the experimental model in the text, i.e. HAP1 WT and dBLM cells complemented with BLM variants, and present FACS profiles of HAP1 variants to confirm comparable ploidy levels in all. As the authors noted in their reply, WT HAP1 cells are expected to be haploid, but they are susceptible to becoming diploid. This must be properly monitored (PMID: 33184093).

We thank the reviewer for the suggestion. However, we did not find the rationale behind how ploidy states rather than the lack of BLM phosphorylation may lead to the premature activation of BLM during mitosis. The key reason we hypothesised that mitotic phosphorylation can suppress BLM activities comes from our important findings of the illegitimate K-chromatin/centromere unwinding triggered by acute PLK1 inhibition. Under such conditions, mitotic BLM is rapidly dephosphorylated and re-localises to the K-chromatin/centromeres. Because of the acute PLK1i treatments (30-60min) and the absence of cell division, the ploidy states/chromosome numbers of a cell do not change. Therefore, it's also hard to conceive that the hyper-activation of mitotic BLM in the HAP1 phospho-mutant cells might be caused by the potential differences of the ploidy state rather than the BLM mutations. Therefore, we do not think the ploidy states are relevant to our study and the FACS results will not change our conclusion. In the above study (PMID: 33184093) mentioned by the reviewer, they were investigating how the cell and nucleus size/shape may change under different ploidy states, which are obviously directly related to DNA contents and chromosome numbers. Therefore, it was reasonable to monitor the ploidy levels.

To completely rule out any possibilities of the mitotic BLM activation that may be caused by secondary or tertiary effects after introducing the BLM phosphorylation mutations, ideally whole genome sequencing and complete cytogenetics, transcriptome and proteome analyses should be

performed in all mutant cells created. This should be the gold standard for any types of research, but unfortunately, this is impractical.

Line 250 – The author should explicitly state that the DNA unwinding activity was 'inferred from centromeric RPA staining' (Ref. 20). While Ref. 20 and Fig. 2E support a correlation between BLM activity and RPA staining, the earlier publication from the same group does not directly substantiate this specific claim. In fact, Ref. 20 uses more cautious wording, stating that “the RPA association likely represents the presence of single-stranded DNA.”

We have shown that a helicase-dead BLM can localise to K-chromatin but is unable to induce RPA foci/threads formation. This is very strong evidence to demonstrate that the centromeric RPA association requires the DNA unwinding action.

The following statement is added to the text: “Since the loading of RPA at both K-chromatin and centromeres relies on active BLM, we thus used RPA foci/threads as an indicator of centromeric DNA unwinding throughout this study.”

Lines 255-256 The sentence “we depleted TOP3A in HeLa and U2OS cells and found that also abolished BLM localisation to UFBs” is unclear. The referent of “that” is ambiguous.

Changed to “we depleted TOP3A in HeLa and U2OS cells and found this also abolished BLM localisation to UFBs in both cell lines.”

Line 262. The authors should justify in the text why they switched to using HeLa cells, i.e. noting that these cells are more resistant to CENP-B overexpression (possibly due to a lack of p53 expression?).

A sentence of “HeLa cells were chosen in these experiments because of the lesser disturbance of the cell cycle progression after the expression of the CENPB-fusion proteins.” was added. We did not want to speculate why HeLa is more resistant than RPE1 cells as there are too many possibilities, which could be due to cancer genome mutations, cell type, viral infection, etc.

Line 290. There is a redundancy in the sentence ‘Repeatedly, we consistently...’

Deleted ‘consistently’

Lines 359-362 - HeLa cells are inherently genetically unstable, leading to substantial variability between cultures across different laboratories (PMID: 26483214, 30778230). Given the inconsistent findings reported in HeLa cells, the author's claim dismissing a role for mitotic BLM in suppressing crossovers appears overstated and should be moderated.

There was only one laboratory using HeLa Δ BLM cells to show the defect of S144A in suppressing SCEs. Data from our lab and other do not find that it fails to suppress SCEs in three other different cell lines. We have mentioned this may be due to cell line/type variations based on the previous suggestion made by this reviewer. Since no other labs have used HeLa cells to repeat the same experiment, we thus will not speculate that the discrepancy may be due to any culture variability across different labs or the unstable genetic makeup of HeLa cells.

In our current study, we have clearly demonstrated that mitotic phosphorylation at S144 and other sites restrict BLM DNA unwinding activity. Moreover, the phosphorylated BLM proteins are actively excluded from DNA/chromatin in mitotic cells. Therefore, it is very hard to imagine how the hyper-

phosphorylated BLM is still competent in processing crossover during mitosis. We thus think that our view is expressed appropriately.

I suggest the authors limit their use of the word “believe”, which appears six times (lines 228, 382, 408, 441, 560, and 129). Its frequent use lends a subjective tone that may detract from the manuscript’s scientific objectivity.

We have minimised the use of the word “believe” according to the reviewer’s suggestion.

Line 228, changed to “speculate”

Line 382, deleted “we believe that”

Line 408, unchanged.

Line 441, unchanged.

Line 560, changed to “think”

Line 129, changed to “was claimed”